# A single gradient step finds adversarial examples on random two-layers neural networks

**Sébastien Bubeck**
Microsoft Research

**Yeshwanth Cherapanamjeri**
UC Berkeley

**Gauthier Gidel**[*]
Mila, Université de Montréal

**Rémi Tachet des Combes**
Microsoft Research

## Abstract

Daniely and Schacham [2020] recently showed that gradient descent finds adversarial examples on random undercomplete two-layers ReLU neural networks. The term "undercomplete" refers to the fact that their proof only holds when the number of neurons is a vanishing fraction of the ambient dimension. We extend their result to the overcomplete case, where the number of neurons is larger than the dimension (yet also subexponential in the dimension). In fact we prove that a single step of gradient descent suffices. We also show this result for any subexponential width random neural network with smooth activation function.

## 1 Introduction

We study the random two-layers neural network model, $f : \mathbb{R}^d \to \mathbb{R}$ defined by

$$f(x) = \frac{1}{\sqrt{k}} \sum_{\ell=1}^{k} a_\ell \psi(w_\ell \cdot x), \tag{1}$$

where $\psi : \mathbb{R} \to \mathbb{R}$ is a fixed non-linearity, the weight vectors $w_\ell \in \mathbb{R}^d$ are i.i.d. from a Gaussian $\mathcal{N}\left(0, \frac{1}{d}\mathrm{I}_d\right)$ (so they are roughly unit norm vectors), and the coefficients $a_\ell \in \mathbb{R}$ are independent from the weight vectors and i.i.d. uniformly distributed in $\{-1, +1\}$. With this parametrization, the central limit theorem says that, for $x \in \sqrt{d} \cdot \mathbb{S}^{d-1}$ (so that $w_\ell \cdot x \sim \mathcal{N}(0, 1)$) and large width $k$, the distribution of $f(x)$ is approximately a centered Gaussian with variance $\mathbb{E}_{X \sim \mathcal{N}(0,1)}[\psi(X)^2]$.

Our goal is to study the concept of *adversarial examples* in this random model. We say that $\delta \in \mathbb{R}^d$ is an *adversarial perturbation* at $x \in \mathbb{R}^d$ if $\|\delta\| \ll \|x\|$ and $\mathrm{sign}(f(x)) \neq \mathrm{sign}(f(x+\delta))$. In this case, we call $x + \delta$ an *adversarial example*. Our main result is that, while $|f(x)| = O(1)$ with high probability, a *single* gradient step on $f$ (i.e., a perturbation of the form $\delta = \eta \nabla f(x)$ for some $\eta \in \mathbb{R}$) suffices to find such adversarial examples, with roughly $\|\delta\| \simeq \frac{\|x\|}{\sqrt{d}} = 1$. Note, here that gradients are taken with respect to the input to the network as opposed to the weights of the network. We prove this statement for networks of subexponential width (e.g., $k \ll \exp(o(d))$) with both smooth and ReLU activation functions. We first state our result for smooth activation functions in the following theorem.

**Theorem 1.** *Let $\gamma \in (0, 1)$ and $\psi$ be non-constant, Lipschitz and with Lipschitz derivative. There exists constants $C_1, C_2, C_3, C_4$ depending on $\psi$ such that the following holds true. Assume $k \geq$*

---

[*]Canada CIFAR AI Chair

$C_1 \log^3(1/\gamma)$ *and* $d \geq C_2 \log(k/\gamma) \log(1/\gamma)$, *and let* $\eta \in \mathbb{R}$ *such that* $|\eta| = C_3 \frac{\sqrt{\log(1/\gamma)}}{\|\nabla f(x)\|^2}$ *and* $\mathrm{sign}(\eta) = -\mathrm{sign}(f(x))$. *Then, with probability at least* $1 - \gamma$, *one has:*

$$\mathrm{sign}(f(x)) \neq \mathrm{sign}(f(x + \eta \nabla f(x))).$$

*Moreover we have* $\|\eta \nabla f(x)\| \leq C_4 \sqrt{\log(1/\gamma)}.$

Note that our proof of Theorem 1 in Section 2 easily gives explicit values for $C_1, C_2, C_3, C_4$. Also note that by re-arranging the constraint on $d$ in Theorem 1 and setting $\gamma = 1/\mathrm{poly}(d)$, the subexponential width condition is of the form $k \ll \exp(o(d))$.

Our second main result establishes similar behavior for the *non-smooth* ReLU activation unit.

**Theorem 2.** *Let* $\gamma \in (0,1)$ *and* $\psi(t) = \max(0, t)$. *There exist constants* $C_1, C_2, C_3, C_4, C_5$ *such that the following holds true. Assume*

$$C_1 \log^6(d) \log(1/\gamma) \leq k, \quad C_2 \log^3(d) \log(1/\gamma) \leq d, \quad C_3 \log^4(k) \log(1/\gamma) \leq \frac{d}{\log(d)}$$

*and let* $\eta \in \mathbb{R}$ *such that* $|\eta| = C_4 \frac{\sqrt{\log(1/\gamma)}}{\|\nabla f(x)\|^2}$ *and* $\mathrm{sign}(\eta) = -\mathrm{sign}(f(x))$. *Then, with probability at least* $1 - \gamma$, *one has:*

$$\mathrm{sign}(f(x)) \neq \mathrm{sign}(f(x + \eta \nabla f(x))).$$

*Moreover, we have* $\|\eta \nabla f(x)\| \leq C_5 \sqrt{\log(1/\gamma)}.$

As before, note that the subexponential condition on the width in the above Theorem is of the form $k \ll \exp(d^{0.24})$. In fact by modifying a bit the proof we can get a condition of the form $k \ll \exp(d^\rho)$ for any $\rho < 1/2$, but for the sake of clarity we only prove the weaker version stated above. The proof for the ReLU activation is broken into two separate cases focusing on the overlapping regimes:

$$\textbf{Case 1: } k \gtrsim d \log^2(d) \quad \text{and} \quad \textbf{Case 2: } \log^6(d) \log(1/\gamma) \lesssim k \lesssim d \log^3 d,$$

as the proofs for the two settings use distinct arguments. The proof for the first regime is similar to the proof for Theorem 1 while the second uses a refinement of an argument by Daniely and Schacham [2020]. These arguments are carried out in Section 3. We would like to note that our results and corresponding proofs are extendible to the setting where $a_\ell$ are drawn from a normal distribution with minor modifications. We include the simpler setting with Bernoulli activations in the last layer for the sake of conceptual clarity. Intuitively, the strong concentration (and anti-concentration) properties of the normal distribution coupled with a conditioning argument on the magnitudes of the weights in the final layer yield similar results for gaussian setting as well. Additionally, due to the scale invariance of the ReLU activation function, our results also hold true for any distribution over $x$ as long as $\mathbb{P}\{x = 0\} = 0$ and for smooth activations when $\mathbb{P}\left\{\|x\| = \Theta(\sqrt{d})\right\} = 1$. Finally, our results also hold when a bias unit is introduced; in this setting, we simply project our perturbation $\delta$ onto the $((d-1)$-dimensional) subspace of perturbations which leave the bias unit unchanged.

## 1.1 Related works

The existence of adversarial examples in neural network architectures was first evidenced in the seminal paper of Szegedy et al. [2014], where the authors found adversarial examples by using the L-BFGS optimization procedure. Shortly after this work, it was hypothesized in Goodfellow et al. [2015] that the existence of adversarial examples stems from an excessive "linearity" of neural network models. This hypothesis was experimentally confirmed by showing that a *single step* of gradient descent suffices to find adversarial perturbations (the so-called *fast gradient sign method* -FGSM). Our theorems can be thought of as a theoretical confirmation of the hypothesis in Goodfellow et al. [2015]. In fact, as explained in Section 1.2, our proofs proceed exactly by showing that "most" two-layers neural networks behave "mostly" linearly over "vast" regions of input space.

We note that not *all* networks are susceptible to one-step gradient attacks to find adversarial examples. Indeed, in Goodfellow et al. [2015], it was shown that adversarial training can be used to build networks that are somewhat robust to one-step gradient attacks. Interestingly, Madry et al. [2018] showed that such models remain susceptible to *multi-steps* gradient attacks, and empirically demonstrated that better robustness can be achieved with adversarial training using multi-steps gradient

attacks. Understanding this phenomenon theoretically remains a challenge, see for example Allen-Zhu and Li [2020] for a proposed approach, and Moosavi-Dezfooli et al. [2019], Qin et al. [2019] for discussion/algorithmic consequences of the relation with the phenomenon of *gradient obfuscation* (Papernot et al. [2017], Athalye et al. [2018]).

Our work is a direct follow-up of Daniely and Schacham [2020] (which itself follows Shamir et al. [2019]). Daniely and Schacham prove that multi-steps gradient descent finds adversarial examples for ReLU random networks of the form (1), as long as the number of neurons is much smaller than the dimension (i.e., $k = o(d)$). They explicitly conjecture that this condition is not necessary, and indeed we exponentially improve it in Theorem 2 (see below for a discussion of $k$ exponential in the dimension). We note that Daniely and Schacham went beyond two-layers neural networks, and conjectured (and proved for shrinking layers) that gradient descent finds adversarial examples on random multi-layers neural networks. We give some experimental confirmation of this multi-layer conjecture in Section 4.

The ultra-wide case $k = \exp(\Omega(d))$ remains open. This exponential size case seems of a different nature than the polynomial size we tackle here, at least for the ReLU activation function. In particular, it is likely that the behavior with exponential width would be closely tied to the actual limit case $k = +\infty$, where the random model (1) yields a Gaussian process. Namely, for $k = +\infty$, $f$ is a Gaussian process indexed by the sphere (say if we restrict to inputs $x \in \sqrt{d} \cdot \mathbb{S}^{d-1}$), with $f(x) \sim \mathcal{N}(0, \mathbb{E}_{X \sim \mathcal{N}(0,1)}[\psi(X)])$ and $\mathbb{E}[f(x)f(y)] = \mathbb{E}_{X,Y \sim \mathcal{N}(0,1):\mathbb{E}[XY]=x \cdot y}[\psi(X)\psi(Y)]$. For example if the activation function is a Hermite polynomial of degree $p$, then $f$ would be a spherical $p$-spin glass model. This polynomial case is particularly well-understood, and in fact the landscape we describe below in Section 1.2 was already described in this case by Ben Arous et al. [2020] (in particular Corollary 59) indicating that adversarial examples would likely continue to persist in the ultra-wide setting. It would be interesting to see if the $p$-spin glass landscape literature can be extended to non-polynomial activation functions, and to a finite (but possibly exponential in $d$) $k$. A step in this latter direction was recently taken in Eldan et al. [2021], where convergence rates to the Gaussian process limit where given both for polynomial activations and for the ReLU. Finally we note that for a smooth activation it might be that there is a more direct argument to remove the subexponential width condition in Theorem 1 (in technical terms, the proof of Proposition 2 could leverage a better argument than our naive upper bound on $\mathrm{Lip}(\Phi)$).

Finally, we note that, in practice, it has been found that there exists "universal" adversarial perturbations that generalize across both inputs and neural networks, Moosavi-Dezfooli et al. [2017]. For the case of ReLU activation (Theorem 2), we could in fact prove our result by replacing the gradient step with a step in the direction $\sum_{\ell=1}^{k} a_\ell w_\ell$, which is indeed a direction *independent* of the input $x$, thus proving the existence of "universal" perturbations (generalizing across inputs) for our model.

## 1.2 The landscape of random two-layers neural networks

For a smooth non-linearity $\psi$, we have

$$\nabla f(x) = \frac{1}{\sqrt{k}} \sum_{\ell=1}^{k} a_\ell w_\ell \psi'(w_\ell \cdot x) \qquad \text{and} \qquad \nabla^2 f(x) = \frac{1}{\sqrt{k}} \sum_{\ell=1}^{k} a_\ell w_\ell w_\ell^\top \psi''(w_\ell \cdot x).$$

We already claimed in the introduction that, with high probability the value of $f(x)$ was bounded:

$$\textbf{Upper-Bounded Value}: \qquad |f(x)| = O(1). \tag{2}$$

We alluded to the CLT for this claim, but it is also easy to guess it intuitively by noting that:

$$\mathbb{E}[f(x)^2] = \mathbb{E}\left[ \frac{1}{k} \sum_{\ell,\ell'=1}^{k} a_\ell a_{\ell'} \psi(w_\ell \cdot x)\psi(w_{\ell'} \cdot x) \right] = \mathbb{E}_{X \sim \mathcal{N}(0,1)}[\psi(X)^2],$$

as $\mathbb{E}[a_\ell a_{\ell'}] = \mathbb{1}\{\ell = \ell'\}$. The formal proof of (2) (and all other claims we make here) will eventually be a simple application of the classical Bernstein concentration inequality. Similarly, it is easy to see that (noticing that $\mathbb{E}[\|\nabla f(x)\|^2] = \mathbb{E}_{X \sim \mathcal{N}(0,1)}[\psi'(X)^2]$), with high probability, the norm of the gradient of $f(x)$ is bounded below (note that we can also show that it is bounded above).

$$\textbf{Lower-Bounded Gradient}: \qquad \|\nabla f(x)\| = \Omega(1). \tag{3}$$

A slightly more difficult calculation, although classical too, is that

$$\textbf{Upper-Bounded Hessian}: \qquad \|\nabla^2 f(x)\|_{\mathrm{op}} = \widetilde{O}\left(\frac{1}{\sqrt{d}}\right). \tag{4}$$

Indeed one can simply note that, for any $u \in \mathbb{S}^{d-1}$, $u^\top \nabla^2 f(x) u = \frac{1}{\sqrt{k}} \sum_{\ell=1}^{k} a_\ell (w_\ell \cdot u)^2 \psi''(w_\ell \cdot x)$ is approximately distributed as a centered Gaussian with variance (through a heuristic application of the asymptotic central limit theorem):

$$\mathbb{E}_{W,Z\sim\mathcal{N}(0,1):\mathbb{E}[WZ]=\frac{x\cdot u}{\sqrt{d}}}\left[\left(\frac{W}{\sqrt{d}}\right)^4 \psi''(Z)^2\right],$$

so that with probability at least $1 - \gamma$ one can expect $u^\top \nabla^2 f(x) u$ to be of order $\frac{\sqrt{\log(1/\gamma)}}{d}$, and thus by taking a union bound over a discretization of the sphere $\mathbb{S}^{d-1}$, one can show inequality (4). In fact, interestingly, one can even hope that (4) holds true for an entire ball around $x$: with appropriate smoothness over $\psi$ (say $C^{2,1}$ smoothness), this could be obtained by another union bound over a second discretization of a $d$-dimensional ball. In other words, we can expect with high probability:

$$\forall x \in \mathbb{R}^d : \|x\| = \mathrm{poly}(d), \text{ one has } \|\nabla^2 f(x)\|_{\mathrm{op}} = \widetilde{O}\left(\frac{1}{\sqrt{d}}\right). \tag{5}$$

Equations (2), (3), and (5) paint a rather clear geometric picture. There are essentially two scales around a fixed $x \in \sqrt{d} \cdot \mathbb{S}^{d-1}$: The *macroscopic scale*, where one considers a perturbation $x + \delta$ with $\|\delta\| = \Omega(\sqrt{d})$, and the *mesoscopic scale* where $\|\delta\| = o(\sqrt{d})$ (we use this term because for the ReLU network there will also be a *microscopic scale*, with $\|\delta\| = o(1)$). At the macroscopic scale the landscape of $f$ might be very complicated, but our crucial observation is that the picture at the mesoscopic scale is dramatically simpler. Namely, at the mesoscopic scale, the function $f$ is essentially linear, since one has (thanks to (3) and (5))

$$\textbf{Approximate Linearity}: \qquad \|\nabla f(x) - \nabla f(x + \delta)\| = o(\|\nabla f(x)\|), \forall \delta : \|\delta\| = o(\sqrt{d}). \tag{6}$$

Moreover, since the height of the function is at most a constant (by (2)) and the norm of the gradient is constant, it suffices to step at a constant distance in the direction of the gradient (or negative gradient) to change the sign of $f$. Assuming without any loss of generality that $f(x) > 0$, we combine (2), (3), and (6) using a standard descent lemma (Lemma 3):

$$f(x - \eta \nabla f(x)) \leq \underbrace{f(x)}_{O(1) \text{ by } (2)} - \eta \underbrace{\|\nabla f(x)\|}_{\Omega(1) \text{ by } (3)} \left( \underbrace{\|\nabla f(x)\|}_{\Omega(1) \text{ by } (3)} - \underbrace{\sup_{\frac{\|\delta\|}{\|\nabla f(x)\|} \leq \eta} \|\nabla f(x) - \nabla f(x + \delta)\|}_{o(1) \text{ by } (4) \text{ or } (6)} \right) \tag{7}$$

$$\leq C_1 - C_2 \eta < 0 \quad (\text{for } C_1/C_2 < \eta < C_3\sqrt{d}) \tag{8}$$

where $C_1, C_2$ and $C_3$ are constants that *do not* depend on $k$ and $d$. In words: a single step of gradient descent (or ascent) with a $O(1)$ step-size suffices to find an adversarial example, and moreover the adversarial perturbation $\delta$ satisfies $\|\delta\| = O(1) = O(\|x\|/\sqrt{d})$.

## 1.3 Proof strategy

The starting point of the proof for both the smooth and ReLU case is to show (2) and (3), which we essentially do below in Section 1.4. In the smooth case, one could then prove formally (4) and conclude as indicated in the last paragraph of Section 1.2. Of course, (4) is simply ill-defined for the ReLU case, so one has to take a different route. Instead, we propose to directly prove (6), that is we study the difference of gradients at the mesoscopic scale. Using that $\|h\| = \sup_{v\in\mathbb{S}^{d-1}} v \cdot h$, we thus need to control (for some $R = o(\sqrt{d})$):

$$\sup_{\delta\in\mathbb{R}^d:\|\delta\|\leq R} \|\nabla f(x) - \nabla f(x + \delta)\| = \sup_{\substack{v\in\mathbb{S}^{d-1}, \\ \delta\in\mathbb{R}^d:\|\delta\|\leq R}} \frac{1}{\sqrt{k}} \sum_{\ell=1}^{k} a_\ell (w_\ell \cdot v)(\psi'(w_\ell \cdot x) - \psi'(w_\ell \cdot (x + \delta))).$$

$$\tag{9}$$

We execute this strategy first for the smooth case in Section 2. We then prove the ReLU case in Section 3, where we face an extraneous difficulty since the gradient is *not* Lipschitz at very small scale, which introduces a third scale (the *microscopic scale*) that has to be dealt with differently. Technically, this issue appears when we try to move from the discretization over $v$ and $\delta$ in (9) to the whole space (a so-called $\varepsilon$-net argument).

## 1.4 Scaling of value and gradient

Here we show how to prove (2) and (3) (in fact, for our purpose, we only need the one-sided inequality $\|\nabla f(x)\| = \Omega(1)$) under very mild conditions on $\psi$ which will be satisfied for both ReLU and smooth activations. We will repeatedly use Bernstein's inequality which we restate here for convenience (see e.g., Theorem 2.10 in Boucheron et al. [2013]):

**Theorem 3** (Bernstein's inequality). *Let $(X_\ell)$ be i.i.d. centered random variables such that for all integers $q \geq 2$, $\mathbb{E}[|X_\ell|^q] \leq \frac{q!}{2}\sigma^2 c^{q-2}$ for fixed $\sigma, c > 0$. Then, with probability at least $1 - \gamma$:*

$$\sum_{\ell=1}^{k} X_\ell \leq \sqrt{2\sigma^2 k \log(1/\gamma)} + c \log(1/\gamma). \tag{10}$$

We will also use repeatedly that $\mathbb{E}_{X\sim\mathcal{N}(0,1)}[|X|^q] \leq (q-1)!! \leq \frac{q!}{2}$, as well as the following concentration of $\chi^2$ random variables (see e.g., (2.19) in Wainwright [2019]): let $X_1, \ldots, X_k$ be i.i.d. standard Gaussians, then with probability at least $1 - \gamma$, one has:

$$\left| \sum_{\ell=1}^{k} X_\ell^2 - k \right| \leq 4\sqrt{k \log(2/\gamma)}. \tag{11}$$

We can now proceed to our various results.

**Lemma 1** (Bounded Value). *Assume that there exists $\sigma, c > 0$ such that for all integers $q \geq 2$, $\mathbb{E}_{X\sim\mathcal{N}(0,1)}[|\psi(X)|^q] \leq \frac{q!}{2}\sigma^2 c^{q-2}$. Then with probability at least $1 - \gamma$ one has*

$$|f(x)| \leq \sqrt{2\sigma^2 \log(1/\gamma)} + \frac{c \log(1/\gamma)}{\sqrt{k}}.$$

*Proof sketch.* Since $\frac{1}{\sqrt{k}} \sum_{\ell=1}^{k} a_\ell \psi(w_\ell \cdot x)$, we use Bernstein's inequality on $X_\ell = a_\ell \psi(w_\ell \cdot x)$. $\square$

**Lemma 2** (Lower-Bounded Gradient). *Let $\psi$ be differentiable almost everywhere, and assume that there exists $\sigma', c' > 0$ such that for all integers $q \geq 2$, $\mathbb{E}_{X\sim\mathcal{N}(0,1)}[|\psi'(X)|^{2q}] \leq \frac{q!}{2}\sigma'^2 c'^{q-2}$. Then with probability at least $1 - \gamma$,*

$$\|\nabla f(x)\| \geq \left( \mathbb{E}_{X\sim\mathcal{N}(0,1)}[|\psi'(X)|^2] - \left( \sqrt{\frac{2\sigma'^2 \log(2/\gamma)}{k}} + \frac{c' \log(2/\gamma)}{k} \right) \right)^{1/2} \left( 1 - 5\sqrt{\frac{\log(4/\gamma)}{d}} \right).$$

*Proof sketch.* Let $P = \mathrm{I}_d - \frac{xx^\top}{d}$ be the projection on the orthogonal complement of the span of $x$. We have $\|\nabla f(x)\| \geq \|P\nabla f(x)\| = \|\frac{1}{\sqrt{k}} \sum_{\ell=1}^{k} P a_\ell w_\ell \psi'(w_\ell \cdot x)\|$ where $a_\ell P w_\ell$ is independent of $w_\ell \cdot x$ and is distributed as $\mathcal{N}\left(0, \frac{1}{d}\mathrm{I}_{d-1}\right)$. We conclude by conditioning on the values $(w_\ell \cdot x)_{\ell\in[k]}$ and using the concentration results (10) and (11). $\square$

## 2 Smooth Non-Linearity (Theorem 1)

In this section, we consider a 1-Lipschitz and $L$-smooth activation function, that is for all $s, t \in \mathbb{R}$,

$$|\psi(s) - \psi(t)| \leq |s - t| \text{ and } |\psi'(s) - \psi'(t)| \leq L|s - t|. \tag{12}$$

We also assume $\psi(0) = 0$ and denote $c_\psi^2 = \mathbb{E}_{X\sim\mathcal{N}(0,1)}[(\psi'(X))^2]$ which we assume to be non-zero (that is $\psi$ is not a constant function).

**Proposition 1** (Upper-Bounded Value and Lower-Bounded Gradient). *Under the above assumptions, one has with probability at least $1 - \gamma$,*

$$|f(x)| \leq \sqrt{2\log\left(\frac{1}{\gamma}\right)} \left(1 + \sqrt{\frac{\log\left(\frac{2}{\gamma}\right)}{k}}\right)$$

$$and \quad \|\nabla f(x)\| \geq \left(c_\psi^2 - \sqrt{\frac{2\log\left(\frac{4}{\gamma}\right)}{k}} \left(1 + \sqrt{\frac{\log\left(\frac{4}{\gamma}\right)}{k}}\right)\right)^{1/2} \left(1 - 5\sqrt{\frac{\log\left(\frac{8}{\gamma}\right)}{d}}\right).$$

*Particularly, there exists $C > 0$ such that for $k \geq C\log(2/\gamma)$ and $d \geq C\log(8/\gamma)$ we have*

$$|f(x)| \leq 2\sqrt{\log(1/\gamma)} \qquad and \qquad \|\nabla f(x)\| \geq c_\psi/2. \tag{13}$$

*Proof.* With (12) we have $|\psi(X)| \leq |X|$ and thus in Lemma 1 we can take $\sigma = c = 1$ which yields the first claimed equation. For the second equation we use that $|\psi'(X)| \leq 1$ (since $\psi$ is 1-Lipschitz) and thus, in Lemma 2, we can also take $\sigma' = c' = 1$ yielding the second claimed equation. $\qquad\square$

Next, we need to control (9) where we use crucially the smoothness of the activation function.

**Proposition 2** (Bounded Variations). *Let $R \geq 1$. With probability at least $1 - \gamma$ one has*

$$\sup_{\delta \in \mathbb{R}^d : \|\delta\| \leq R} \|\nabla f(x) - \nabla f(x + \delta)\| \leq 20RL\left(\sqrt{\frac{\log(Rk/\gamma)}{d}} + \frac{\log(1/\gamma)}{\sqrt{k}}\right).$$

*Particularly, for any $c > 0$, there exists $C_1, C_2$ such that if $k \geq C_1 R^2 log^2(1/\gamma)$ and $d \geq C_2 R^2 \log(Rk/\gamma)$ then*

$$\sup_{\delta \in \mathbb{R}^d : \|\delta\| \leq R} \|\nabla f(x) - \nabla f(x + \delta)\| \leq c. \tag{14}$$

*Proof sketch.* Our plan is to use the fact that $\|h\| = \sup_{v \in \mathbb{S}^{d-1}} v \cdot h$. Thus, we first show, by using Bernstein's inequality on $X_\ell = \frac{a_\ell}{L}(w_\ell \cdot v)(\psi'(w_\ell \cdot x) - \psi'(w_\ell \cdot (x + \delta)))$, that if we fix $\delta \in \mathbb{R}^d$ such that $\|\delta\| \leq R$ and $v \in \mathbb{S}^{d-1}$, then with probability at least $1 - \gamma$ one has:

$$\Phi(v, \delta) := \langle \nabla f(x) - \nabla f(x + \delta), v \rangle = \frac{L}{\sqrt{k}} \sum_{\ell=1}^{k} X_l \leq \frac{4RL}{d}\sqrt{\log(1/\gamma)}\left(1 + \sqrt{\frac{\log(1/\gamma)}{k}}\right).$$

Let $\Omega := \{(v, \delta) : \|v\| = 1, \|\delta\| \leq R\}$ and $N_\varepsilon$ be an $\varepsilon$-net over $\Omega$ with $\varepsilon = 1/k$. By a union bound over $N_\varepsilon$ (whose size is at most $(10kR)^{2d}$ [Vershynin, 2018, Corollary 4.2.13]), we obtain with probability at least $1 - \gamma$:

$$\sup_{(v,\delta) \in \Omega} \Phi(v, \delta) \leq \sup_{(v,\delta) \in N_\varepsilon} \Phi(v, \delta) + \sup_{(v,\delta),(v',\delta') \in \Omega : \|v-v'\| + \|\delta - \delta'\| \leq \varepsilon} |\Phi(v, \delta) - \Phi(v', \delta')|$$

$$\leq \frac{4RL}{d}\sqrt{2d\log(Rk) + \log(1/\gamma)}\left(1 + \sqrt{\frac{2d\log(Rk) + \log(1/\gamma)}{k}}\right) + \frac{\mathrm{Lip}(\Phi)}{k}.$$

Then we show that the variations of $\Phi$ can be upper-bounded by a $RL$ times a $\chi^2$ random variable and use (11) to show that with probability at least $1 - \gamma$,

$$\mathrm{Lip}(\Phi) \leq RL\left(\sqrt{k} + 4\sqrt{\frac{\log(1/\gamma)}{d}}\right). \qquad\square$$

Finally we can turn to the proof of Theorem 1. Let us first recall this theorem.

**Theorem 1.** *Let $\gamma \in (0,1)$ and $\psi$ be non-constant, Lipschitz and with Lipschtiz derivative. There exists constants $C_1, C_2, C_3, C_4$ depending on $\psi$ such that the following holds true. Assume $k \geq C_1 \log^3(1/\gamma)$ and $d \geq C_2 \log(k/\gamma) \log(1/\gamma)$, and let $\eta \in \mathbb{R}$ such that $|\eta| = C_3 \sqrt{\log(1/\gamma)} \|\nabla f(x)\|^{-2}$ and $\mathrm{sign}(\eta) = -\mathrm{sign}(f(x))$. Then with probability at least $1 - \gamma$ one has:*

$$\mathrm{sign}(f(x)) \neq \mathrm{sign}(f(x + \eta \nabla f(x))).$$

*Moreover we have $\|\eta \nabla f(x)\| \leq C_4 \sqrt{\log(1/\gamma)}$.*

*Proof.* We make the following claims which hold with probability at least $1 - \gamma$. Without any loss of generality we can assume that $f(x) > 0$, we will use a standard descent lemma (Lemma 3) and the previous propositions to get that,

$$f(x - \eta \nabla f(x)) \leq \underbrace{f(x)}_{O(1) \text{ by } (13)} - \eta \underbrace{\|\nabla f(x)\|}_{\Omega(1) \text{ by } (13)} \left( \underbrace{\|\nabla f(x)\|}_{\Omega(1) \text{ by } (13)} - \underbrace{\sup_{\frac{\|\delta\|}{\|\nabla f(x)\|} \leq \eta} \|\nabla f(x) - \nabla f(x + \delta)\|}_{o(1) \text{ by } (14)} \right). \quad (15)$$

Formally, let us set $\eta = \frac{32}{c_\psi^2 \|\nabla f(x)\|^2} \sqrt{\log(1/\gamma)}$, $R = \frac{64}{c_\psi^3} \sqrt{\log(1/\gamma)}$, $k \geq C_1 R^2 \log^2(1/\gamma)$ and $d \geq C_2 R^2 \log(Rk/\gamma)$ where $C_1$ and $C_2$ are large enough such that (13) and (14) are valid with $c = \frac{c_\psi}{4}$. By Proposition 1 we have that $\|\nabla f(x)\| \geq c_\psi/2$ and $\eta \|\nabla f(x)\|^2 = \frac{32}{c_\psi^2} \sqrt{\log(1/\gamma)} \geq \frac{16}{c_\psi^2} |f(x)|$. Moreover, Proposition 2 shows that for all $\delta$ such that $\|\delta\| \leq |\eta| \|\nabla f(x)\| \leq \frac{64}{c_\psi^3} \sqrt{\log(1/\gamma)} = R$, we thus have $\|\nabla f(x) - \nabla f(x + \delta)\| \leq c := \frac{c_\psi}{4}$. Consequently,

$$f(x - \eta \nabla f(x)) \leq f(x) - \eta \|\nabla f(x)\| \left( \frac{c_\psi}{2} - \frac{c_\psi}{4} \right) \leq f(x) - \eta \frac{c_\psi^2}{8} \leq -f(x). \quad (16)$$

Thus, with a single gradient step of size at most $R = \frac{64}{c_\psi^3} \sqrt{\log(1/\gamma)}$, we switched the sign of $f$. $\quad \square$

## 3  ReLU Non-Linearity (Theorem 2)

In this section, we consider the ReLU non-linearity $\psi(t) = \max(0, t)$. We start by showing that $f(x)$ is upper bounded and that $\|\nabla f(x)\|$ is lower-bounded.

**Proposition 3** (Upper-Bounded Value and Lower-Bounded Gradient). *With probability at least $1 - \gamma$,*

$$|f(x)| \leq \sqrt{2 \log(2/\gamma)} \left( 1 + \sqrt{\frac{\log(2/\gamma)}{k}} \right)$$

$$and \quad \|\nabla f(x)\| \geq \left( \frac{1}{2} - \sqrt{\frac{2 \log(4/\gamma)}{k}} \left( 1 + \sqrt{\frac{\log(1/\gamma)}{k}} \right) \right)^{1/2} \left( 1 - 5\sqrt{\frac{\log(4/\gamma)}{d}} \right).$$

*Particularly, there exists $C > 0$ such that for $k \geq C \log(1/\gamma)$ and $d \geq C \log(4/\gamma)$ we have*

$$|f(x)| \leq 2\sqrt{\log(2/\gamma)} \quad and \quad \|\nabla f(x)\| \geq \frac{1}{4}. \quad (17)$$

*Proof.* In Lemma 1 and Lemma 2, we can take $\sigma = c = \sigma' = c' = 1$ (since $|\psi(X)| \leq |X|$ and $|\psi'(X)| \leq 1$), which concludes the proof. $\quad \square$

Now we split the control of the gradient variation into two cases: the large width case ($Cd \log^2 d \lesssim k$) and the small width case ($\log^6(d) \log(1/\gamma) \lesssim k \lesssim d \log^3 d$).

**Proposition 4.** *(Bounded Variations – Large Width Case) Let $1 \leq R \leq \sqrt{d}/2$, $\sqrt{k} \geq 52$ and $d \geq \log(1/\gamma)$. Then, with probability at least $1 - \gamma$, one has*

$$\sup_{\delta \in \mathbb{R}^d : \|\delta\| \leq R} \|\nabla f(x) - \nabla f(x + \delta)\| \leq 20 \left( R \log^2(Rk) \sqrt{\frac{\log d}{d}} \right)^{1/4} + 40 \sqrt{\frac{d}{k}} \log(Rk).$$

*Particularly, for any $c > 0$, there exists $C_1, C_2$ such that if $C_1 d \log^2(Rk) \le k$ and $C_2 \log^4(Rk)R^2 \le \frac{d}{\log(d)}$, then we have,*

$$\sup_{\delta \in \mathbb{R}^d \,:\, \|\delta\| \le R} \|\nabla f(x) - \nabla f(x + \delta)\| \le c. \tag{18}$$

*Proof sketch.* In the smooth case (Proposition 2) we did so by using crucially the smoothness of the activation function. Here, instead of smoothness, we will use that only few activations can change when you make *microscopic move* (i.e., between $x + \delta$ and $x + \delta'$ with $\|\delta - \delta'\| = o(1)$). The key observation that for any $\delta$ such that $\|\delta\| \le R$,

$$\mathbb{P}(\mathrm{sign}(w_\ell \cdot x) \ne \mathrm{sign}(w_\ell \cdot (x + \delta))) \le R\sqrt{\frac{2\log(d)}{d}} + \frac{1}{d}. \tag{19}$$

We can then use this result to bound the variation of $\nabla f(x)$ along a fixed direction. If we fix $\delta \in \mathbb{R}^d$ such that $\|\delta\| \le R$ (with $R \ge 1$) and $v \in \mathbb{S}^{d-1}$, then with probability at least $1 - \gamma$ one has:

$$\Phi(v, \delta) := \langle \nabla f(x) - \nabla f(x + \delta), v \rangle = \frac{1}{\sqrt{k}} \sum_{\ell=1}^{k} X_l \le 2\sqrt{\frac{\log(\frac{1}{\gamma})}{d}} \left( \left(2R\sqrt{\frac{\log(d)}{d}}\right)^{\frac{1}{4}} + \sqrt{\frac{\log(\frac{1}{\gamma})}{k}} \right).$$

This inequality is proven via Bernstein's inequality on $X_\ell := a_\ell(w_\ell \cdot v)(\psi'(w_\ell \cdot x) - \psi'(w_\ell \cdot (x + \delta)))$ where we use (19) to prove

$$\mathbb{E}[|X_\ell|^q] \le \sqrt{\mathbb{E}[|w_\ell \cdot v|^{2q}] \times \mathbb{P}(\mathrm{sign}(w_\ell \cdot x) \ne \mathrm{sign}(w_\ell \cdot (x + \delta)))} \le \frac{q!}{2} c^{q-2} \times \sigma^2, \tag{20}$$

with $\sigma = \frac{2}{\sqrt{d}} \times \left(2R\sqrt{\frac{\log(d)}{d}}\right)^{1/4}$ and $c = \frac{2}{\sqrt{d}}$.

Then, similarly as in the proof of Proposition 2, we use a covering argument (but this time with an $\varepsilon$-Net of size $\varepsilon = R^{-4/3}k^{-4}$) to show that,

$$\sup_{(v,\delta) \in \Omega} \Phi(v, \delta) \le \sup_{(v,\delta) \in N_\varepsilon} \Phi(v, \delta) + \sup_{(v,\delta),(v',\delta') \in \Omega : \|v-v'\| + \|\delta - \delta'\| \le \varepsilon} |\Phi(v, \delta) - \Phi(v', \delta')|$$

$$\le 2\sqrt{\frac{10d\log(Rk) + \log(2/\gamma)}{d}} \left( \left(2R\sqrt{\frac{\log(d)}{d}}\right)^{1/4} + \sqrt{\frac{10d\log(Rk) + \log(2/\gamma)}{k}} \right)$$

$$+ \sup_{(v,\delta),(v',\delta') \in \Omega : \|v-v'\| + \|\delta - \delta'\| \le \varepsilon} |\Phi(v, \delta) - \Phi(v', \delta')|. \tag{21}$$

By using $\chi^2$ concentration, it is easy to see that with probability at least $1 - \gamma$:

$$|\Phi(\delta, v) - \Phi(\delta, v')| \le \|v - v'\| \sqrt{k + 4k\sqrt{\frac{\log(k/\gamma)}{d}}}. \tag{22}$$

By using (19) we are able to show that with probability $1 - \delta$ for any $(v, \delta) \in N_\varepsilon$ there is *at most* $4d$ different activated neurons between $\Phi(v, \delta)$ and $\Phi(v, \delta')$ and thus by applying a concentration result of Lipschitz function of Gaussians we get,

$$|\Phi(\delta, v) - \Phi(\delta', v)| \le 18\sqrt{\frac{d}{k}} \sqrt{\log 4k + \frac{\log 8/\gamma}{d}}. \tag{23}$$

Combining (21), (22), and (23), we get

$$\sup_{\delta \in \mathbb{R}^d \,:\, \|\delta\| \le R} \|\nabla f(x) - \nabla f(x + \delta)\| \le 20 \left( R \log^2(Rk) \sqrt{\frac{\log d}{d}} \right)^{1/4} + 40\sqrt{\frac{d}{k}} \log(Rk). \quad \square$$

**Proposition 5** (Bounded Variations – Small Width Case). *Let $20 \log^3 d \cdot \log 1/\gamma \le k \le d \log^3 d$ and $d \ge 20 \cdot \log^3 d \cdot \log 1/\gamma$. Then, with probability at least $1 - \gamma$, we have:*

$$\sup_{\delta \in \mathbb{R}^d, \|\delta\| \le \frac{\sqrt{k}}{\log^6 d}} \|\nabla f(x) - \nabla f(x + \delta)\| \le \frac{60}{\log d}.$$

*Proof sketch.* In the small width regime we assume $20 \log^3 d \cdot \log 1/\delta \le k \le d \log^3 d$. The idea of the proof is to control the operator norm of $W$ (where $f(x) = \frac{1}{\sqrt{k}} a^\top \psi(W^\top x)$) as well as the number of neurons with a too small activation. With high probability, we show that we have,

$$\|W\| \le 3 \cdot \left(1 + \sqrt{\frac{10(k + \log 1/\delta)}{d}}\right), \quad \# \left\{i : |\langle w_i, x \rangle| \le \frac{1}{\log^3 d}\right\} \le 2 \cdot \frac{k}{\log^3 d}.$$

Thus, uniformly on $\left\{y : \|x - y\| \le \frac{\sqrt{k}}{\log^6 d}\right\}$ the cardinal of $T_y := \{i : \operatorname{sgn}\langle w_i, x \rangle \ne \operatorname{sgn}\langle w_i, y \rangle\}$ is uppper-bounded:

$$|T_y| \le 300 k / \log^3 d := B. \tag{24}$$

We can conclude the proof by using an union bound and a concentration inequality on Gaussians.

$$\|\nabla f(x) - \nabla f(y)\| = \frac{1}{\sqrt{k}} \left\|\sum_{\ell \in T} a_\ell w_\ell\right\| \le \sup_{\substack{A \subset [k] \\ |A| \le B}} \frac{1}{\sqrt{k}} \left\|\sum_{\ell \in T} a_\ell w_\ell\right\| \le \sqrt{\frac{B}{k}} \left(1 + \sqrt{\frac{B \log k + \log \frac{4}{\delta}}{d}}\right).$$

$\square$

We can now prove our final result.

**Theorem 2.** *Let $\gamma \in (0,1)$ and $\psi(t) = \max(0, t)$. There exist constants $C_1, C_2, C_3, C_4, C_5$ such that the following holds true. Assume*

$$C_1 \log^6(d) \log(1/\gamma) \le k, \quad C_2 \log^3(d) \log(1/\gamma) \le d, \quad C_3 \log^4(k) \log(1/\gamma) \le \frac{d}{\log(d)},$$

*and let $\eta \in \mathbb{R}$ such that $|\eta| = C_4 \frac{\sqrt{\log(1/\gamma)}}{\|\nabla f(x)\|^2}$ and $\operatorname{sign}(\eta) = -\operatorname{sign}(f(x))$. Then with probability at least $1 - \gamma$ one has:*

$$\operatorname{sign}(f(x)) \ne \operatorname{sign}(f(x + \eta \nabla f(x))).$$

*Moreover we have $\|\eta \nabla f(x)\| \le C_5 \sqrt{\log(1/\gamma)}$.*

*Proof.* The proof is the same as for Theorem 1 using Proposition 3 instead of Proposition 1 and Proposition 4 or Proposition 5 replacing Proposition 2. $\square$

## 4 Experiments

**Setting.** To verify our theoretical findings, we run some experiments to measure empirically the probability of finding an adversarial example in the direction $\nabla f(x)$. More precisely, we take a random point $x$ of norm $\sqrt{d}$ and initialize a network using the procedure described in Section 1[2]. We then find the smallest $\eta$ such that a gradient step $\eta \nabla f(x)$ changes the sign of the function. $\eta$ is of the opposite sign of $f(x)$ and we limit our search to $|\eta| < 20$. We explore various values of $d$ and $k$, as well as deeper networks with $L = 1$ through $L = 6$ hidden layers. All hidden layers are of width $k$. For our experiments, we used ReLU activation units and the standard deviations reported in Figure 1 are computed over 100 random network initializations for each choice of architecture and 100 random input points for each such initialization.

**Results.** Figure 2a shows the average of the smallest $\eta$ required to switch the sign of the function. We note that the average only includes cases where an $\eta$ was indeed found. Figure 2b shows the gradient norm in $x$ (all cases included). As we see, both the smallest $\eta$ and the gradient norm are approximately constant both in $d$ and in $k$. This finding also holds for deeper networks (see Appendix D). In Figure 1, we show the fraction of examples (out of $10,000$ samples) whose sign is switched. We see that with $L = 1$ and values of $d$ and $k$ larger than 50, $100\%$ of samples are switched. This confirms our theoretical results. Once again, we observe the same statement holds for deeper networks. The values of $d$ and $k$ at which $100\%$ switching is reached appears to grow with $L^3$. Additionally, we track the value of the smallest $\eta$ required to switch the sign of the function, and the gradient norm w.r.t $x$. Both are approximately constant in $d$ and in $k$. This finding also holds for deeper networks (see Appendix D for the curves).

---

[2]We also conducted experiments with a gaussian initialization for the last layer in place of the Bernoulli initialization from Section 1 and obtained qualitatively similar results.

[3]Due to GPU memory limitations, $k$ could not reach 1,000,000 for deeper networks.

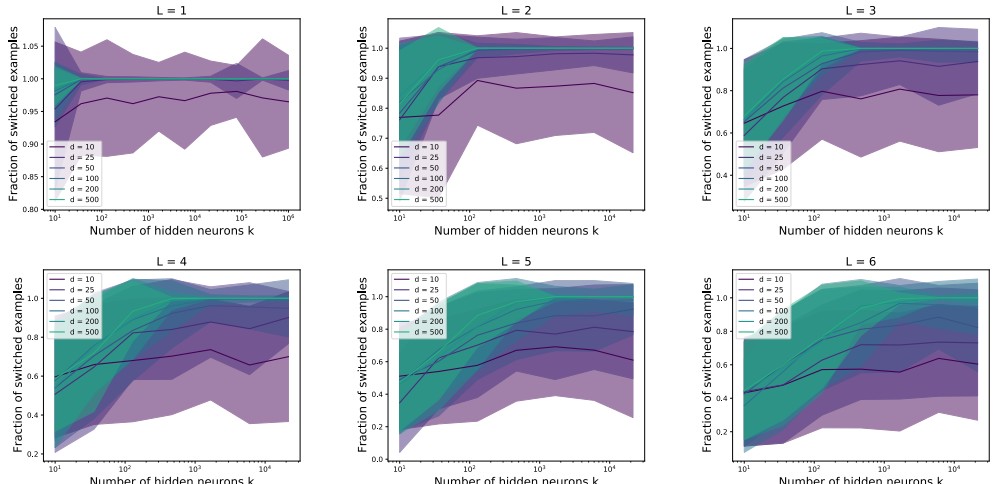

**Figure 1:** Fraction of inputs with an adversarial example found after a single gradient step, for various input dimensions $d$, hidden layer widths $k$ and number of hidden layers $L$. For each pair $(k, L)$, we report the average over 100 network initializations and 100 values of $x$ per initialization. The colored area represents one standard deviation.

### Acknowledgment

We thank Mark Sellke for pointing out to us the reference Ben Arous et al. [2020], and Peter Bartlett for several discussions on this problem.

## 5    Disclosure of Funding and Competing Interests

GG research is supported by the Canada CIFAR AI Chair Program and an IVADO grant. YC gratefully acknowledges the support of the NSF through grants DMS-2023505 and DMS-2031883, the Simons Foundation through award #814639, and Microsoft through the BAIR Open Research Commons. We declare no competing interests.

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
