# A  Proofs for Section 1 (Introduction)

**Lemma 3.** *For any continuous and almost everywhere differentiable function $f$, and any $x \in \mathbb{R}^d$ and $\eta \in \mathbb{R}$, one has:*

$$\left| f\left( x + \frac{\eta}{\|\nabla f(x)\|^2} \nabla f(x) \right) - (f(x) + \eta) \right| \leq \sup_{\delta \in \mathbb{R}^d : \|\delta\| \leq \frac{\eta}{\|\nabla f(x)\|}} |\eta| \frac{\|\nabla f(x) - \nabla f(x+\delta)\|}{\|\nabla f(x)\|} .$$

*Proof.* Let $g(t) = f\left( x + t\frac{\eta}{\|\nabla f(x)\|^2} \nabla f(x) \right)$ so that

$$
\begin{aligned}
g'(t) &= \frac{\eta}{\|\nabla f(x)\|^2} \nabla f(x) \cdot \nabla f\left( x + t\frac{\eta}{\|\nabla f(x)\|^2} \nabla f(x) \right) \\
&= \eta + \eta \frac{\nabla f(x)}{\|\nabla f(x)\|} \cdot \frac{\nabla f\left( x + t\frac{\eta}{\|\nabla f(x)\|^2} \nabla f(x) \right) - \nabla f(x)}{\|\nabla f(x)\|} .
\end{aligned}
$$

Thus we have:

$$|g(1) - g(0) - \eta| \leq \int_0^1 |g'(t) - \eta| dt \leq |\eta| \int_0^1 \frac{\left\| \nabla f\left( x + t\frac{\eta}{\|\nabla f(x)\|^2} \nabla f(x) \right) - \nabla f(x) \right\|}{\|\nabla f(x)\|} dt ,$$

which concludes the proof. $\square$

**Lemma 1** (Bounded Value). *Assume that there exists $\sigma, c > 0$ such that for all integers $q \geq 2$, $\mathbb{E}_{X \sim \mathcal{N}(0,1)}[|\psi(X)|^q] \leq \frac{q!}{2} \sigma^2 c^{q-2}$. Then with probability at least $1 - \gamma$ one has*

$$|f(x)| \leq \sqrt{2\sigma^2 \log(1/\gamma)} + \frac{c \log(1/\gamma)}{\sqrt{k}} .$$

*Proof.* Let us recall that,

$$f(x) = \frac{1}{\sqrt{k}} \sum_{\ell=1}^k a_\ell \psi(w_\ell \cdot x) , \tag{25}$$

Let $X_\ell = a_\ell \psi(w_\ell \cdot x)$. Then $\mathbb{E}[X_\ell] = 0$ and

$$\mathbb{E}[|X_\ell|^q] \leq \frac{q!}{2} \sigma^2 c^{q-2}, \text{ for all integers } q \geq 2 .$$

Thus Bernstein's inequality states that with probability at least $1 - \gamma$ one has

$$\sqrt{k} f(x) = \sum_{\ell=1}^k X_\ell \leq \sqrt{2\sigma^2 k \log(1/\gamma)} + c \log(1/\gamma) .$$

$\square$

**Lemma 2** (Lower-Bounded Gradient). *Let $\psi$ be differentiable almost everywhere, and assume that there exists $\sigma', c' > 0$ such that for all integers $q \geq 2$, $\mathbb{E}_{X \sim \mathcal{N}(0,1)}[|\psi'(X)|^{2q}] \leq \frac{q!}{2} \sigma'^2 c'^{q-2}$. Then with probability at least $1 - \gamma$,*

$$\|\nabla f(x)\| \geq \left( \mathbb{E}_{X \sim \mathcal{N}(0,1)}[|\psi'(X)|^2] - \left( \sqrt{\frac{2\sigma'^2 \log(2/\gamma)}{k}} + \frac{c' \log(2/\gamma)}{k} \right) \right)^{1/2} \left( 1 - 5\sqrt{\frac{\log(4/\gamma)}{d}} \right) .$$

*Proof.* Let us first recall that

$$\nabla f(x) = \frac{1}{\sqrt{k}} \sum_{\ell=1}^k a_\ell w_\ell \psi'(w_\ell \cdot x) \tag{26}$$

We start with a first technical lemma.

**Lemma 4.** *Let $\psi$ be differentiable almost everywhere. Then with probability at least $1 - \gamma$ for $0 < \gamma < 2/e$ one has:*

$$\|\nabla f(x)\| \geq \left(1 - 5\sqrt{\frac{\log(2/\gamma)}{d}}\right)\sqrt{\frac{1}{k}\sum_{\ell=1}^{k}\psi'(w_\ell \cdot x)^2}.$$

*Proof.* Let $P = \mathrm{I}_d - \frac{xx^\top}{d}$ be the projection on the orthogonal complement of the span of $x$. We have $\|\nabla f(x)\| \geq \|P\nabla f(x)\|$. Moreover $a_\ell P w_\ell$ is independent of $w_\ell \cdot x$, and thus conditioning on the values $(w_\ell \cdot x)_{\ell \in [k]}$ we obtain (using that $a_\ell P w_\ell$ is distributed as $\mathcal{N}\left(0, \frac{1}{d}\mathrm{I}_{d-1}\right)$):

$$P\nabla f(x) = \frac{1}{\sqrt{k}}\sum_{\ell=1}^{k} a_\ell P w_\ell \psi'(w_\ell \cdot x) \stackrel{(d)}{=} \left(\sqrt{\frac{1}{kd}\sum_{\ell=1}^{k}\psi'(w_\ell \cdot x)^2}\right) Y \text{ where } Y \sim \mathcal{N}\left(0, \mathrm{I}_{d-1}\right).$$

Using (11) we have that with probability at least $1 - \gamma$:

$$\|Y\|^2 \geq d - 1 - 4\sqrt{d\log(2/\gamma)} \geq d - 5\sqrt{d\log(2/\gamma)}.$$

where we used that $d \geq 1$ and $\gamma < 2/e$. The two above displays easily conclude the proof. $\qquad\square$

A straightforward application of Bernstein's inequality to random variables $\psi'(w_\ell \cdot x)^2$ (which are $O(1)$-sub-exponential from our smoothness assumptions on $\psi$) yields with probability at least $1 - \gamma$:

$$\frac{1}{k}\sum_{\ell=1}^{k}\psi'(w_\ell \cdot x)^2 \geq \mathbb{E}_{X\sim\mathcal{N}(0,1)}[|\psi'(X)|^2] - \left(\sqrt{\frac{2\sigma'^2\log(1/\gamma)}{k}} + \frac{c'\log(1/\gamma)}{k}\right).$$

It suffices to combine this inequality with Lemma 4 and apply a direct union bound. $\qquad\square$

# B Proofs for Section 2 (Smooth Non-Linearity (Theorem 1))

**Proposition 2** (Bounded Variations). *Let $R \geq 1$. With probability at least $1 - \gamma$ one has*

$$\sup_{\delta \in \mathbb{R}^d : \|\delta\| \leq R} \|\nabla f(x) - \nabla f(x + \delta)\| \leq 20RL\left(\sqrt{\frac{\log(Rk/\gamma)}{d}} + \frac{\log(1/\gamma)}{\sqrt{k}}\right).$$

*Particularly, for any $c > 0$, there exists $C_1, C_2$ such that if $k \geq C_1 R^2 \log^2(1/\gamma)$ and $d \geq C_2 R^2 \log(Rk/\gamma)$ then*

$$\sup_{\delta \in \mathbb{R}^d : \|\delta\| \leq R} \|\nabla f(x) - \nabla f(x + \delta)\| \leq c. \tag{14}$$

*Proof.* We will crutially use the fact that for any $h \in \mathbb{R}^d$ we have.

$$\|h\| = \sup_{v \in \mathbb{S}^{d-1}} v \cdot h \tag{27}$$

We first start with a lemma for fixed $v$ and $\delta$.

**Lemma 5.** *Fix $\delta \in \mathbb{R}^d$ such that $\|\delta\| \leq R$ and $v \in \mathbb{S}^{d-1}$. Then with probability at least $1 - \gamma$ one has:*

$$\langle \nabla f(x) - \nabla f(x + \delta), v \rangle \leq \frac{4RL}{d}\sqrt{\log(1/\gamma)}\left(1 + \sqrt{\frac{\log(1/\gamma)}{k}}\right).$$

*Proof.* Let us recall that

$$\langle \nabla f(x) - \nabla f(x + \delta), v \rangle = \frac{1}{\sqrt{k}}\sum_{\ell=1}^{k} a_\ell(w_\ell \cdot v)(\psi'(w_\ell \cdot x) - \psi'(w_\ell \cdot (x + \delta))) \tag{28}$$

We apply Bernstein's inequality with $X_\ell = \frac{a_\ell}{L}(w_\ell \cdot v)(\psi'(w_\ell \cdot x) - \psi'(w_\ell \cdot (x+\delta)))$. We have $\mathbb{E}[X_\ell] = 0$ and (by $L$-smoothness of $\psi$)

$$
\begin{aligned}
\mathbb{E}[|X_\ell|^q] \leq \mathbb{E}[|w_\ell \cdot v|^q |w_\ell \cdot \delta|^q] &\leq \sqrt{\mathbb{E}[|w_\ell \cdot v|^{2q}]\mathbb{E}[|w_\ell \cdot \delta|^{2q}]} \\
&= \frac{\|\delta\|^q}{d^q}\mathbb{E}_{X \sim \mathcal{N}(0,1)}[|X|^{2q}] \leq (2q-1)!!\left(\frac{R}{d}\right)^q \leq \frac{q!}{2}\left(\frac{2R}{d}\right)^q.
\end{aligned}
$$

Thus we can apply Bernstein with $\sigma = c = \frac{2R}{d}$ which yields the claimed bound. $\qquad\square$

Now, we can prove our main result.

Denote $\Phi(v,\delta) = \frac{1}{\sqrt{k}}\sum_{\ell=1}^k a_\ell(w_\ell \cdot v)(\psi'(w_\ell \cdot x) - \psi'(w_\ell \cdot (x+\delta)))$. In Lemma 5, we controlled $\Phi(v,\delta)$ for a fixed $v$ and $\delta$. We now want to control it uniformly over $\Omega = \{(v,\delta) : \|v\| = 1, \|\delta\| \leq R\}$. To do so, we apply an union bound over an $\varepsilon$-net for $\Omega$, denote it $N_\varepsilon$, whose size is then at most $(10R/\varepsilon)^{2d}$ [Vershynin, 2018, Corollary 4.2.13]. In particular; we obtain with probability at least $1 - \gamma$:

$$
\sup_{(v,\delta)\in\Omega}\Phi(v,\delta) \leq \sup_{(v,\delta)\in N_\varepsilon}\Phi(v,\delta) + \sup_{(v,\delta),(v',\delta')\in\Omega:\|v-v'\|+\|\delta-\delta'\|\leq\varepsilon}|\Phi(v,\delta) - \Phi(v',\delta')|
$$

$$
\leq \frac{4RL}{d}\sqrt{2d\log(10R/\varepsilon) + \log(1/\gamma)}\left(1 + \sqrt{\frac{2d\log(10R/\varepsilon) + \log(1/\gamma)}{k}}\right) \quad (29)
$$

$$
+ \varepsilon \times \mathrm{Lip}(\Phi). \quad (30)
$$

Thus, it only remains to estimate the Lipschitz constant of the mapping $\Phi$. To do so, note that for any $\delta, \delta'$,

$$
|\Phi(\delta,v) - \Phi(\delta',v)| \leq \frac{L\|\delta - \delta'\|}{\sqrt{k}}\sum_{\ell=1}^k\|w_\ell\|^2,
$$

and similarly for any $v, v'$,

$$
|\Phi(\delta,v) - \Phi(\delta,v')| \leq \frac{RL\|v - v'\|}{\sqrt{k}}\sum_{\ell=1}^k\|w_\ell\|^2.
$$

Using (11), we have with probability at least $1 - \gamma$ that

$$
\sum_{\ell=1}^k\|w_\ell\|^2 \leq k + 4\sqrt{\frac{k\log(1/\gamma)}{d}}. \quad (31)
$$

Thus with see that with probability at least $1 - \gamma$,

$$
\mathrm{Lip}(\Phi) \leq RL\left(\sqrt{k} + 4\sqrt{\frac{\log(1/\gamma)}{d}}\right).
$$

Combining this with (30) concludes the proof (by taking $\varepsilon = 1/k$ and with straightforward algebraic manipulations). $\qquad\square$

## C   Proofs for Section 3 (ReLU Non-Linearity (Theorem 2))

**Proposition 3** (Upper-Bounded Value and Lower-Bounded Gradient). *With probability at least $1 - \gamma$,*

$$
|f(x)| \leq \sqrt{2\log(2/\gamma)}\left(1 + \sqrt{\frac{\log(2/\gamma)}{k}}\right)
$$

*and* $\quad \|\nabla f(x)\| \geq \left(\frac{1}{2} - \sqrt{\frac{2\log(4/\gamma)}{k}}\left(1 + \sqrt{\frac{\log(1/\gamma)}{k}}\right)\right)^{1/2}\left(1 - 5\sqrt{\frac{\log(4/\gamma)}{d}}\right).$

*Particularly, there exists $C > 0$ such that for $k \geq C\log(1/\gamma)$ and $d \geq C\log(4/\gamma)$ we have*

$$
|f(x)| \leq 2\sqrt{\log(2/\gamma)} \qquad and \qquad \|\nabla f(x)\| \geq \frac{1}{4}. \quad (17)
$$

**Proposition 4.** *(Bounded Variations – Large Width Case) Let* $1 \leq R \leq \sqrt{d}/2$, $\sqrt{k} \geq 52$ *and* $d \geq \log(1/\gamma)$. *Then, with probability at least* $1 - \gamma$, *one has*

$$\sup_{\delta \in \mathbb{R}^d \,:\, \|\delta\| \leq R} \|\nabla f(x) - \nabla f(x + \delta)\| \leq 20 \left( R \log^2(Rk) \sqrt{\frac{\log d}{d}} \right)^{1/4} + 40 \sqrt{\frac{d}{k}} \log(Rk).$$

*Particularly, for any $c > 0$, there exists $C_1, C_2$ such that if $C_1 d \log^2(Rk) \leq k$ and $C_2 \log^4(Rk) R^2 \leq \frac{d}{\log(d)}$, then we have,*

$$\sup_{\delta \in \mathbb{R}^d \,:\, \|\delta\| \leq R} \|\nabla f(x) - \nabla f(x + \delta)\| \leq c. \tag{18}$$

*Proof.* In the smooth case we did so via Proposition 2, which both used crucially the smoothness of the activation function. Here, instead of smoothness, we will use that only few activations can change when you make *microscopic move* (i.e., between $x + \delta$ and $x + \delta'$ with $\|\delta - \delta'\| = o(1)$). The key observation is the following lemma:

**Lemma 6.** *For any $\delta$ such that $\|\delta\| \leq R$,*

$$\mathbb{P}(\text{sign}(w_\ell \cdot x) \neq \text{sign}(w_\ell \cdot (x + \delta))) \leq R \sqrt{\frac{2 \log(d)}{d}} + \frac{1}{d}. \tag{32}$$

*Moreover, for any $\delta$ with $\|\delta\| \leq \sqrt{d}/2$, we have*

$$\mathbb{P}(\exists \delta' : \|\delta - \delta'\| \leq \varepsilon \text{ and } \text{sign}(w_\ell \cdot (x + \delta)) \neq \text{sign}(w_\ell \cdot (x + \delta'))) \leq 2\varepsilon \left( 1 + 2 \sqrt{\frac{\log(2/\varepsilon)}{d}} \right). \tag{33}$$

*Proof.* We have:

$$\mathbb{P}(\text{sign}(w_\ell \cdot x) \neq \text{sign}(w_\ell \cdot (x + \delta))) \leq \mathbb{P}(|w_\ell \cdot \delta| \geq |w_\ell \cdot x|) \leq \mathbb{P}(|w_\ell \cdot \delta| \geq t) + \mathbb{P}(|w_\ell \cdot x| \leq t),$$

where the last inequality holds for any threshold $t \in \mathbb{R}$. Now, note that $w_\ell \cdot \delta \sim \mathcal{N}(0, \frac{\|\delta\|^2}{d})$ and $w_\ell \cdot x \sim \mathcal{N}(0, 1)$. Thus picking $t = R\sqrt{\frac{2\log(d)}{d}}$ and applying standard tail bounds on Gaussian random variables for the first term in the above expression and the fact that the pdf of a standard gaussian is upper bounded by $1/\sqrt{2\pi}$ for the second, shows that

$$\mathbb{P}(\text{sign}(w_\ell \cdot x) \neq \text{sign}(w_\ell \cdot (x + \delta))) \leq R \sqrt{\frac{2\log(d)}{d}} + \frac{1}{d},$$

which concludes the proof of (32).

For (33) we have:

$$\begin{aligned}
&\mathbb{P}(\exists \delta' : \|\delta - \delta'\| \leq \varepsilon \text{ and } \text{sign}(w_\ell \cdot (x + \delta)) \neq \text{sign}(w_\ell \cdot (x + \delta'))) \\
&\leq \mathbb{P}(\exists \delta' : \|\delta - \delta'\| \leq \varepsilon \text{ and } |w_\ell \cdot (\delta' - \delta)| \geq t) + \mathbb{P}(|w_\ell \cdot (x + \delta)| \leq t) \\
&\leq \mathbb{P}(\|w_\ell\| \geq t/\varepsilon) + \mathbb{P}(|w_\ell \cdot (x + \delta)| \leq t).
\end{aligned}$$

where $w_\ell \cdot (x + \delta) \sim \mathcal{N}(0, \sigma^2)$ with $\sigma^2 \geq \frac{1}{2}$ since $\|\delta\| \leq \sqrt{d}/2$. Thus picking $t = \varepsilon \sqrt{1 + 4\sqrt{\frac{\log(2/\varepsilon)}{d}}}$ and applying (11) (as $\|w_\ell\|^2$ is a $\chi^2$ random variable) concludes the proof. $\square$

We will crucially use the fact that for any $h \in \mathbb{R}^d$ we have.

$$\|h\| = \sup_{v \in \mathbb{S}^{d-1}} v \cdot h \tag{34}$$

We first start with a lemma for fixed $v$ and $\delta$ which is the equivalent of Lemma 5:

**Lemma 7.** *Fix $\delta \in \mathbb{R}^d$ such that $\|\delta\| \leq R$ (with $R \geq 1$) and $v \in \mathbb{S}^{d-1}$. Then with probability at least $1 - \gamma$ one has:*

$$\frac{1}{\sqrt{k}} \sum_{\ell=1}^{k} a_\ell (w_\ell \cdot v)(\psi'(w_\ell \cdot x) - \psi'(w_\ell \cdot (x + \delta))) \leq 2 \sqrt{\frac{\log(1/\gamma)}{d}} \left( \left( 2R \sqrt{\frac{\log(d)}{d}} \right)^{1/4} + \sqrt{\frac{\log(1/\gamma)}{k}} \right).$$

*Proof.* We apply Bernstein's inequality with $X_\ell = a_\ell(w_\ell \cdot v)(\psi'(w_\ell \cdot x) - \psi'(w_\ell \cdot (x + \delta)))$. We have $\mathbb{E}[X_\ell] = 0$ and (using (32) in Lemma 6)

$$
\begin{aligned}
\mathbb{E}[|X_\ell|^q] &= \mathbb{E}[|w_\ell \cdot v|^q |\psi'(w_\ell \cdot x) - \psi'(w_\ell \cdot (x + \delta))|^q] \\
&\leq \sqrt{\mathbb{E}[|w_\ell \cdot v|^{2q}] \times \mathbb{P}(\mathrm{sign}(w_\ell \cdot x) \neq \mathrm{sign}(w_\ell \cdot (x + \delta)))} \\
&\leq \sqrt{\frac{(2q)!}{2d^q}} \times \sqrt{2R\sqrt{\frac{\log(d)}{d}}} \\
&\leq \frac{q!}{2}\left(\frac{2}{\sqrt{d}}\right)^q \times \sqrt{2R\sqrt{\frac{\log(d)}{d}}} \, .
\end{aligned}
$$

Thus we can apply Bernstein with $\sigma = \frac{2}{\sqrt{d}} \times \left(2R\sqrt{\frac{\log(d)}{d}}\right)^{1/4}$ and $c = \frac{2}{\sqrt{d}}$ which yields the claimed bound. $\qquad\square$

Similarly to the proof of Proposition 2, we define $\Phi(v, \delta) = \frac{1}{\sqrt{k}}\sum_{\ell=1}^{k} a_\ell(w_\ell \cdot v)(\psi'(w_\ell \cdot x) - \psi'(w_\ell \cdot (x + \delta)))$, and $N_\varepsilon$ an $\varepsilon$-net for $\Omega = \{(v, \delta), \|v\| = 1, \|\delta\| \leq R\}$ (recall that $|N_\varepsilon| \leq (10R/\varepsilon)^{2d}$). Using Lemma 7, we obtain with probability at least $1 - \gamma$:

$$
\sup_{(v,\delta)\in\Omega} \Phi(v,\delta) \leq \sup_{(v,\delta)\in N} \Phi(v,\delta) + \sup_{(v,\delta),(v',\delta')\in\Omega:\|v-v'\|+\|\delta-\delta'\|\leq\varepsilon} |\Phi(v,\delta) - \Phi(v',\delta')|
$$

$$
\leq 2\sqrt{\frac{2d\log(10R/\varepsilon) + \log(1/\gamma)}{d}} \left(\left(2R\sqrt{\frac{\log(d)}{d}}\right)^{1/4} + \sqrt{\frac{2d\log(10R/\varepsilon) + \log(1/\gamma)}{k}}\right)
$$

$$
+ \sup_{(v,\delta),(v',\delta')\in\Omega:\|v-v'\|+\|\delta-\delta'\|\leq\varepsilon} |\Phi(v,\delta) - \Phi(v',\delta')| \, . \tag{35}
$$

Thus, it remains again to estimate the scale of the oscillations of the mapping $\Phi$ but crucially only *at scale $\varepsilon$* (the crucial point is that we don't need to argue about infinitesimal scale, where a ReLU network is *not* smooth). For $v, v'$, one has

$$
|\Phi(\delta, v) - \Phi(\delta, v')| \leq \frac{\|v - v'\|}{\sqrt{k}} \sum_{\ell=1}^{k} \|w_\ell\| \, .
$$

Using (11), we see that with probability at least $1 - \gamma$, one has for all $\ell \in [k]$,

$$
\|w_\ell\|^2 \leq 1 + 4\sqrt{\frac{\log(k/\gamma)}{d}} \, ,
$$

so that in this event we have:

$$
|\Phi(\delta, v) - \Phi(\delta, v')| \leq \|v - v'\|\sqrt{k + 4k\sqrt{\frac{\log(k/\gamma)}{d}}} \, . \tag{36}
$$

On the other hand, for $\delta, \delta'$ we write:

$$
\begin{aligned}
|\Phi(\delta, v) - \Phi(\delta', v)| &\leq \frac{1}{\sqrt{k}}\left|\sum_{\ell=1}^{k} \mathbb{1}\{\mathrm{sign}(w_\ell \cdot (x + \delta)) > \mathrm{sign}(w_\ell \cdot (x + \delta'))\} a_\ell w_\ell \cdot v\right| \\
&+ \frac{1}{\sqrt{k}}\left|\sum_{\ell=1}^{k} \mathbb{1}\{\mathrm{sign}(w_\ell \cdot (x + \delta)) < \mathrm{sign}(w_\ell \cdot (x + \delta'))\} a_\ell w_\ell \cdot v\right| \\
&\leq \frac{1}{\sqrt{k}}\left\|\sum_{\ell=1}^{k} \mathbb{1}\{\mathrm{sign}(w_\ell \cdot (x + \delta)) > \mathrm{sign}(w_\ell \cdot (x + \delta'))\} a_\ell w_\ell\right\| \\
&+ \frac{1}{\sqrt{k}}\left\|\sum_{\ell=1}^{k} \mathbb{1}\{\mathrm{sign}(w_\ell \cdot (x + \delta)) < \mathrm{sign}(w_\ell \cdot (x + \delta'))\} a_\ell w_\ell\right\| \tag{37}
\end{aligned}
$$

Letting $X_\ell(\delta) = \mathbb{1}\{\exists \delta' : \|\delta - \delta'\| \leq \varepsilon \text{ and } \operatorname{sign}(w_\ell \cdot (x + \delta)) \neq \operatorname{sign}(w_\ell \cdot (x + \delta'))\}$, we now control with exponentially high probability $\sum_{\ell=1}^{k} X_\ell(\delta)$. By (33) in Lemma 6, we know that $X_\ell(\delta)$ is a Bernoulli of parameter at most $2\varepsilon \left(1 + 2\sqrt{\frac{\log(2/\varepsilon)}{d}}\right)$. So we have:

$$\mathbb{P}\left(\sum_{\ell=1}^{k} X_\ell(\delta) \geq s\right) \leq \left(2k\varepsilon\left(1 + 2\sqrt{\frac{\log(2/\varepsilon)}{d}}\right)\right)^s.$$

And thus, thanks to an union bound, we obtain:

$$\mathbb{P}\left(\exists (v, \delta) \in N_\varepsilon : \sum_{\ell=1}^{k} X_\ell(\delta) \geq s\right) \leq \left(\frac{10R}{\varepsilon}\right)^{2d}\left(2k\varepsilon\left(1 + 2\sqrt{\frac{\log(2/\varepsilon)}{d}}\right)\right)^s. \tag{38}$$

With $s = 4d$ the latter is upper bounded by $(26k\sqrt{R}\varepsilon^{3/8})^{4d}$ (using the fact that $\sqrt{\varepsilon}(1 + 2\sqrt{\log 2/\varepsilon}) \leq 4\varepsilon^{3/8}$, $\forall 1 \geq \varepsilon > 0$). Taking $\varepsilon = R^{-4/3}k^{-4}$ we get that this probability is less than $(26/\sqrt{k})^{8d} \leq \gamma$ for $\sqrt{k} \geq 52$ and $d \geq \log(1/\gamma)$.

Furthermore, we have by another union bound and the concentration of Lipschitz functions of Gaussians [Boucheron et al., 2013, Theorem 5.5] ($\|\cdot\|$ is a 1-Lipschitz function):

$$\mathbb{P}\left(\exists S \subset [k], |S| \leq 4d : \left\|\frac{1}{\sqrt{k}}\sum_{i \in S} a_i w_i\right\| \geq \sqrt{\frac{|S|}{k}}(1 + t)\right) \leq k^{4d}e^{-\frac{dt^2}{2}}$$

By setting $t = 2\sqrt{\log 4k + \frac{\log 8/\gamma}{d}}$, we get that with probability at least $1 - \gamma/8$:

$$\forall S \subset [k], |S| \leq 4d : \left\|\frac{1}{\sqrt{k}}\sum_{i \in S} a_i w_i\right\| \leq 9\sqrt{\frac{d}{k}}\sqrt{\log 4k + \frac{\log 8/\gamma}{d}} \tag{39}$$

Finally, noting that for all $(v, \delta) \in N, \|\delta' - \delta\| \leq \varepsilon$:

$$\mathbb{1}\{\operatorname{sign}(w_\ell \cdot (x + \delta)) < \operatorname{sign}(w_\ell \cdot (x + \delta'))\} \leq X_\ell(\delta)$$
$$\mathbb{1}\{\operatorname{sign}(w_\ell \cdot (x + \delta)) > \operatorname{sign}(w_\ell \cdot (x + \delta'))\} \leq X_\ell(\delta),$$

we may combine (36), (37), (38) and (39) to obtain that with probability at least $1 - \gamma$, we have for all $(\delta, v) \in N_\varepsilon$ and $\delta', v'$ with $\|\delta - \delta'\| \leq \frac{1}{R^{4/3}k^4}$ and $\|v - v'\| \leq \frac{1}{R^{4/3}k^4}$,

$$|\Phi(\delta, v) - \Phi(\delta, v')| \leq \frac{1}{R^{4/3}k^4}\sqrt{k + 4k\sqrt{\frac{\log(4k/\gamma)}{d}}} .$$

and

$$|\Phi(\delta, v) - \Phi(\delta', v)| \leq 18\sqrt{\frac{d}{k}}\sqrt{\log 4k + \frac{\log 8/\gamma}{d}} .$$

Combining this with (35) we obtain with probability at least $1 - \gamma$:

$$\sup_{(v, \delta) \in \Omega} \Phi(v, \delta)$$
$$\leq 2\sqrt{\frac{10d\log(Rk) + \log(2/\gamma)}{d}}\left(\left(2R\sqrt{\frac{\log(d)}{d}}\right)^{1/4} + \sqrt{\frac{10d\log(Rk) + \log(2/\gamma)}{k}}\right)$$
$$+ 20\sqrt{\frac{d}{k}}\sqrt{\log 4k + \frac{\log 8/\gamma}{d}}$$
$$\leq 3\sqrt{\frac{10d\log(Rk) + \log(2/\gamma)}{d}}\left(\left(2R\sqrt{\frac{\log(d)}{d}}\right)^{1/4} + \sqrt{\frac{10d\log(Rk) + \log(2/\gamma)}{k}}\right),$$

which concludes the proof up to straightforward algebraic manipulations. $\square$

**Proposition 5** (Bounded Variations – Small Width Case). *Let $20 \log^3 d \cdot \log 1/\gamma \leq k \leq d \log^3 d$ and $d \geq 20 \cdot \log^3 d \cdot \log 1/\gamma$. Then, with probability at least $1 - \gamma$, we have:*

$$\sup_{\delta \in \mathbb{R}^d, \|\delta\| \leq \frac{\sqrt{k}}{\log^6 d}} \|\nabla f(x) - \nabla f(x + \delta)\| \leq \frac{60}{\log d} .$$

*Proof.* Throughout this proof, we will refer to the weight mapping the input to the network to the inputs to the hidden layers by $W$; that is:

$$f(x) = \frac{1}{\sqrt{k}} a^\top \psi(W^\top x) \quad \text{where} \quad W^\top = \begin{bmatrix} w_1^\top \cdots w_k \top \end{bmatrix} \quad \text{and} \quad a^\top = [a_1 \cdots a_k]$$

We start by proving a simple lemma bounding the spectral norm of $W$:

**Lemma 8.** *We have:*

$$\|W\| \leq 2 \cdot \left( 1 + \sqrt{\frac{10(k + \log 1/\gamma)}{d}} \right)$$

*with probability at least $1 - \gamma$.*

*Proof.* Let $\mathcal{G}$ be a $1/3$-net of $\mathbb{S}^{k-1}$. Then, we have for all $\|v\| = 1$:

$$\|v^\top W\| \leq \|\widetilde{v}^\top W\| + \|(v - \widetilde{v})^\top W\| \leq \|\widetilde{v}^\top W\| + \frac{\|W\|}{3}$$

where $\widetilde{v} = \mathrm{argmin}_{u \in \mathcal{G}} \|v - u\|$. By maximizing over $v$ and re-arranging the inequality, we get:

$$\|W\| \leq \frac{3}{2} \cdot \max_{u \in \mathcal{G}} \|u^\top W\|. \tag{40}$$

Note that we may assume $|\mathcal{G}| \leq (10)^k$ [Vershynin, 2012, Corollary 4.2.3]. For any fixed $u \in \mathcal{G}$, we have that $u^\top W \sim \mathcal{N}(0, I_d/d)$ and hence we have:

$$\|u^\top W\| \leq \left( 1 + \sqrt{\frac{\log 1/\gamma^\dagger}{d}} \right)$$

with probability at least $1 - \gamma^\dagger$. By setting $\gamma^\dagger = \gamma/|\mathcal{G}|$ and a union bound over $\mathcal{G}$, we have with probability at least $1 - \gamma$:

$$\forall u \in \mathcal{G} : \|u^\top W\| \leq \left( 1 + \sqrt{\frac{10(k + \log 1/\gamma)}{d}} \right) .$$

The conclusion now follows from (40). $\qquad\square$

We prove a lemma which will restrict how many neurons change their activation patterns in a ball around $x$.

**Lemma 9.** *For $k \geq 10 \cdot \log^3 d \cdot \log 1/\gamma$, we have that:*

$$\# \left\{ i : |\langle x, w_i \rangle| \leq \frac{1}{\log^3 d} \right\} \leq 2 \cdot \frac{k}{\log^3 d}$$

*with probability at least $1 - \gamma$.*

*Proof.* We have that $\langle w_i, x \rangle \sim \mathcal{N}(0, 1)$ and from the fact that the density of a standard gaussian random variable is bounded by $1/\sqrt{2\pi}$, we get:

$$\mathbb{P} \left\{ |\langle w_i, x \rangle| \leq \frac{1}{\log^3 d} \right\} \leq \frac{1}{\log^3 d} .$$

Since $\langle w_i, x \rangle$ are mutually independent, the conclusion now follows from [Mitzenmacher and Upfal, 2017, Theorem 4.4]. $\qquad\square$

We first instantiate Lemmas 8 and 9 with failure probabilities set to $\gamma/4$ in each. Therefore, we have:

$$\|W\| \leq 3 \cdot \left(1 + \sqrt{\frac{10(k + \log 1/\gamma)}{d}}\right), \quad \#\left\{i : |\langle w_i, x\rangle| \leq \frac{1}{\log^3 d}\right\} \leq 2 \cdot \frac{k}{\log^3 d}.$$

Let $R = \frac{\sqrt{k}}{\log^6 d}$, by using $20\log^3 d \cdot \log 1/\gamma \leq k \leq d\log^3 d$ we have:

$$\forall y \text{ s.t } \|y - x\| \leq R : \|W(y - x)\| \leq 15R\log^{3/2} d. \tag{41}$$

Now, let $\|y - x\| \leq R$, $S = \left\{i : |\langle w_i, x\rangle| \leq \frac{1}{\log^3 d}\right\}$ and $T = \{i : \text{sgn}\langle w_i, x\rangle \neq \text{sgn}\langle w_i, y\rangle\}$. We have:

$$\forall i \in T \setminus S : |\langle w_i, y - x\rangle| \geq \frac{1}{\log^3 d}.$$

Hence, we get:

$$\sqrt{|T \setminus S|} \cdot \frac{1}{\log^3 d} \leq \|W(y - x)\| \leq 15R\log^{3/2} d \implies |T \setminus S| \leq 250 \cdot \frac{k}{\log^3 d}.$$

Therefore, along with our bound on $|S|$, we have:

$$|T| \leq 300 \cdot \frac{k}{\log^3 d} =: L.$$

To conclude the proof, let $A \subset [k]$ with $|A| \leq L$. We now have:

$$Y_A := \frac{1}{\sqrt{k}} \sum_{\ell \in A} a_\ell w_\ell \sim \mathcal{N}\left(0, \frac{|A|}{k} \cdot \frac{I_d}{d}\right) \implies \|Y_A\| \leq \sqrt{\frac{L}{k}} \cdot \left(1 + \sqrt{\frac{\log 1/\gamma^\dagger}{d}}\right)$$

with probability at least $1 - \gamma^\dagger$. Setting $\gamma^\dagger = \gamma/(4 \cdot k^L)$, we have:

$$\forall A \subset [k], |A| \leq L : \|Y_A\| \leq \sqrt{\frac{L}{k}} \cdot \left(1 + \sqrt{\frac{L\log k + \log 4/\gamma}{d}}\right) \leq \frac{30}{\log d}$$

with probability at least $1 - \gamma/4$. Recall that:

$$\nabla f(x) - \nabla f(y) = \frac{1}{\sqrt{k}} \cdot \sum_{\ell=1}^{k} a_\ell w_\ell \left(\mathbb{1}\{\text{sign}(w_\ell \cdot x) > 0\} - \mathbb{1}\{\text{sign}(w_\ell \cdot y) > 0\}\right)$$

which along with the previous discussion implies the lemma. $\qquad \square$

## D   Additional Results

We report in Fig. 2-7 the smallest $\eta$ to switch the sign of the prediction and the gradient norm at $x$ for depths $L \in \{1, \ldots, 6\}$. In all our plots, the results are averaged over 100 network initializations and 100 values of $x$ per initialization and the colored area represents one standard deviation. We consider values of $d$ in $[10, 500]$ and of $k$ in $[10, 1e6]$. The code is attached to the submission, we ran it on an internal gpu cluster with P40/P100 cards, but it is simple enough to run on a cpu.

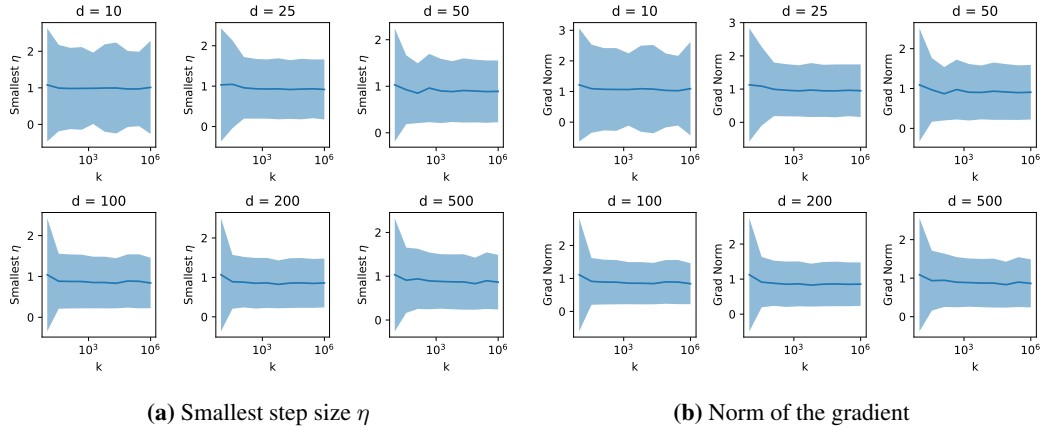

**(a)** Smallest step size $\eta$        **(b)** Norm of the gradient

**Figure 2:** Smallest step-size $\eta$ switching the prediction (**left**) and average gradient norm $\|\nabla f(x)\|$ (**right**) for L = 1. Averages over 100 network initializations and 100 values of $x$ per initialization. The colored area represents one standard deviation.

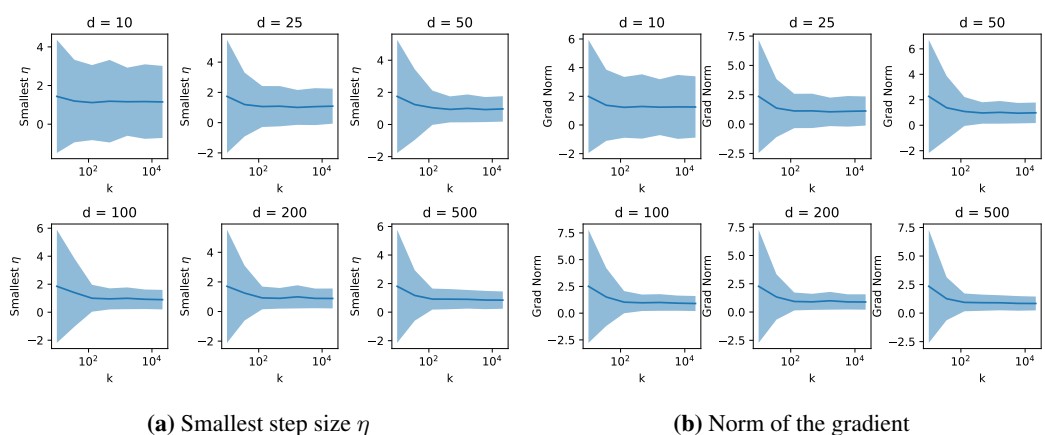

**(a)** Smallest step size $\eta$        **(b)** Norm of the gradient

**Figure 3:** Smallest $\eta$ switching the prediction and average gradient norm $\|\nabla f(x)\|$ for L = 2.

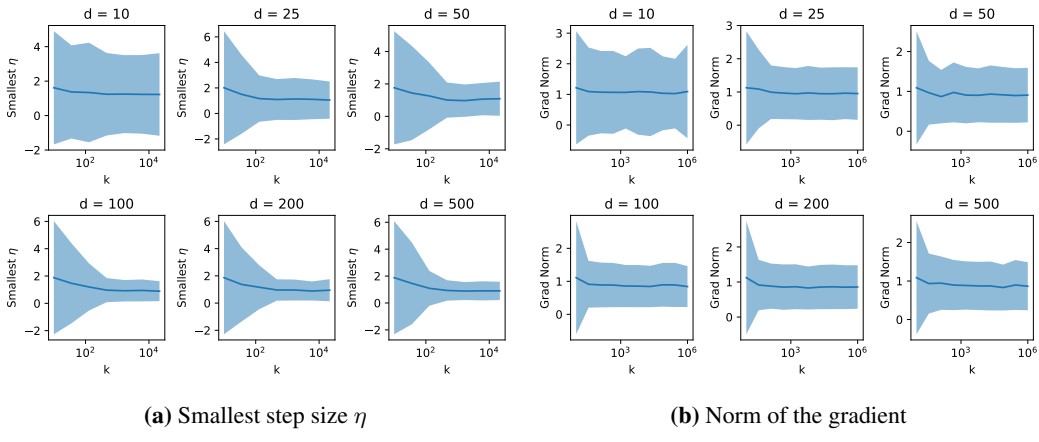

**(a)** Smallest step size $\eta$        **(b)** Norm of the gradient

**Figure 4:** Smallest $\eta$ switching the prediction and average gradient norm $\|\nabla f(x)\|$ for L = 3.

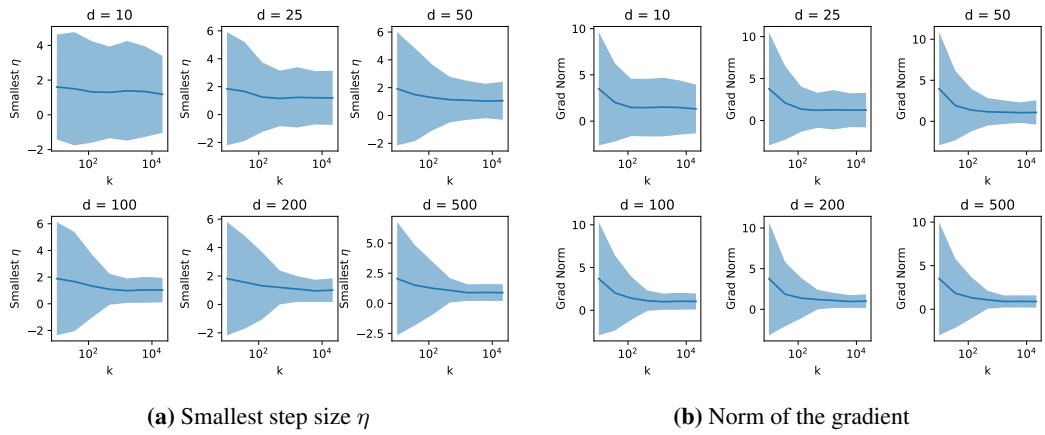

(a) Smallest step size $\eta$  (b) Norm of the gradient

**Figure 5:** Smallest $\eta$ switching the prediction and average gradient norm $\|\nabla f(x)\|$ for L=4.

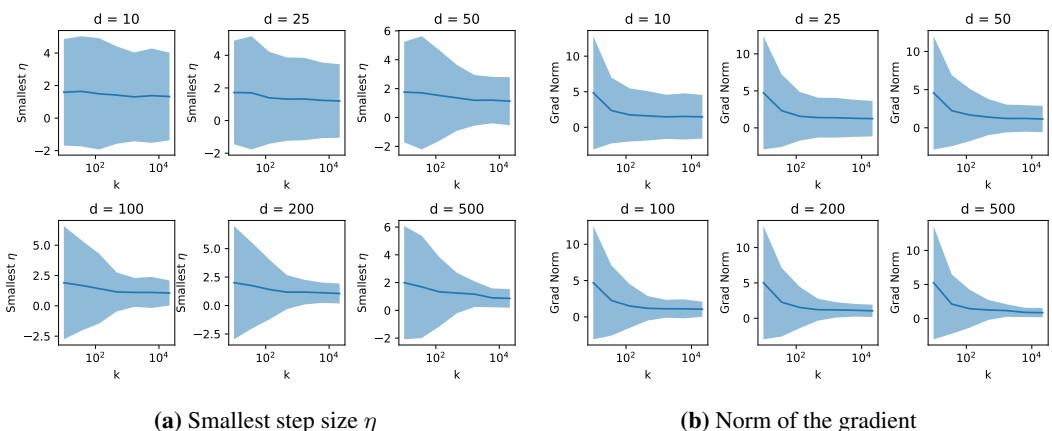

(a) Smallest step size $\eta$  (b) Norm of the gradient

**Figure 6:** Smallest $\eta$ switching the prediction and average gradient norm $\|\nabla f(x)\|$ for L = 5.

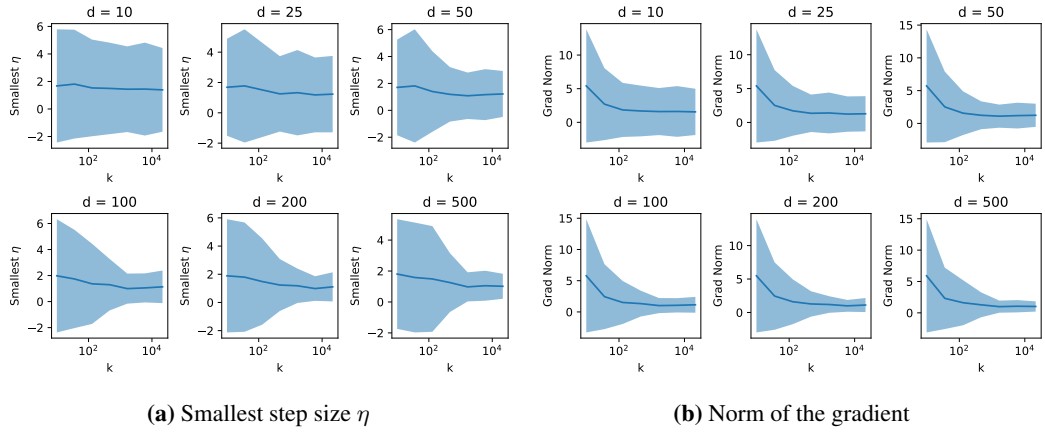

(a) Smallest step size $\eta$  (b) Norm of the gradient

**Figure 7:** Smallest $\eta$ switching the prediction and average gradient norm $\|\nabla f(x)\|$ for L =6.