# OpenReview forum: "A single gradient step finds adversarial examples on random two-layers neural networks"
_NeurIPS.cc/2021/Conference — NeurIPS 2021 Spotlight_

### Official Review · Reviewer_ATH6 · 2021-07-10

**Rating:** 7
**Confidence:** 4

**Summary:**

The paper studies adversarial examples in random depth-2 ReLU networks.  They consider depth-2 networks, where the first layer is drawn from a normal distribution such that the norms of the weights vectors are roughly 1, and the second layer is drawn from the uniform distribution on {-1/\sqrt{k},1/\sqrt{k}} where k is the width of the network. They assume that k is at most sub-exponential in the input dimension, and is bounded below by some polylogarithmic expression. For an input x on the sphere of radius \sqrt{d} the output of the network is w.h.p. O(1) (the input x is fixed, and the probability is over the choice of the network). The authors show that there is w.h.p. \delta with ||\delta||=O(1) such that sign(f(x)) \neq sign(f(x+\delta)). Moreover, such \delta can be found with a single gradient step. Thus, although the inputs are of size \sqrt{d}, there is w.h.p. a perturbation of size O(1) that changes the sign of the output. They prove the result for both smooth and ReLU activations.
Finally, they validate the results empirically, and also demonstrate that ש similar phenomenon occurs in deeper networks.

Daniely and Schacham [2020] showed that such a property holds in ReLU networks where the sizes d_1,…,d_t  of the layers (where d_1 is the input dimension) are such that for every j we have d_{j+1} = o(d_j) and d_t = \omega(1). For the special case of depth-2 networks it required width k=o(d). Thus, the improvement here w.r.t. k is exponential.

The proofs roughly follow the following steps. First, they show an O(1) upper bound on f(x) and an \Omega(1) lower bound on ||\nabla f(x)||. Then, they bound the variation of the gradient near x. Finally, they use standard arguments to bound f(x-\eta \nabla f(x)) for an appropriate \eta. The proofs of these steps for the case of the ReLU activation is different from the case of smooth activations, and the proofs for the ReLU activation with small k is different from the case of large k.


**Limitations And Societal Impact:**

Yes

**Main Review:**

The paper is well-written and the main ideas of the proofs are clearly explained in the first 9 pages. Since real-world trained networks are highly non-random, then I think that the nature of adversarial examples in random networks might be very different than in the case of trained networks. Still, I find the mathematical formulation of the problem on random networks natural, and I think that by understanding adversarial examples in random networks we may gain useful insights. Thus, the setting of random networks is a first step towards more realistic settings. The improvement w.r.t. the results of Daniely and Schacham is significant, since their assumption of k=o(d) is very strong.

The random networks in this paper are such that the second layer is drawn from the uniform distribution on {-1/\sqrt{k},1/\sqrt{k}}. This weights distribution is less standard. I think that a result for the case where all weights are drawn from a normal distribution would be interesting. Can your results be extended to this setting? Anyway, I think that a discussion on this issue is appropriate. Also, in Daniely and Schacham all weights are drawn from a normal distribution, and hence in this sense the results here are not strictly stronger even for depth-2 networks.
I am also not sure what is the initialization scheme in the experiments on deep networks. I guess that all layers are drawn from a normal distribution, except maybe for the final layer. I think that a more detailed description on this setting is required.

A minor comment: In the last equation in page 8, in the expression after the first inequality the sum should be over A.

Overall, I think that the paper gives significant results on a natural problem, and I recommend acceptance.

**post rebuttal update**
Thank you for your response. I maintain my recommendation for acceptance.

**Time Spent Reviewing:**

6

---

> ### Author Response · Authors · 2021-08-10
> **Initial Author Response**
>
> Thanks for your review!
>
> On the extension of our results to a gaussian initialization of the final layer, we reproduce our response to Reviewer ogDp. We believe our results to be easily extendable to the setting where the weights in the final layer are normally distributed. One simple way of obtaining these results is to first condition on the largest magnitude of the $a_\ell$. This is less than $C \sqrt{\log k / \gamma}$ with probability at least $1 - \gamma$. One can then carry out the rest of the proof conditioned on this event. This results in a weaker bound in Proposition 2 where the upper bound incurs an additional $\sqrt{\log k}$ multiplicative factor. Similarly, a lower bound on the size of the gradient follows by exploiting the anti-concentration properties of the Normal distribution using essentially the same proof as in Proposition 1. Here, one simply conditions on the event that at least $k/2$ of the $a_\ell$ have magnitude larger than $1/2$ which happens with exponentially high probability. Subsequently conditioning on the $a_\ell$ and carrying out the rest of the proof of Proposition 1 as is leads to a weaker bound with an additional multiplicative factor of (at least) $1 / 8$ in the lower bound. The upper bound on $|f(x)|$ follows a similar computation. We will include a more comprehensive discussion of this extension in the paper.
>
> Indeed, in the experiments, we used a normal distribution at initialization, except for the last layer which follows our theoretical setting. We also ran some experiments initializing the last layer with a normal and obtained essentially the same curves. We will add a discussion of this.

---

### Official Review · Reviewer_ogDp · 2021-07-16

**Rating:** 7
**Confidence:** 3

**Summary:**

The paper studies the phenomenon of adversarial examples for random two-layer networks with ReLU or smooth activations. It is shown that one step of gradient descent yields adversarial examples (i.e., a switching of the sign of the network output) with high probability, under a much milder condition on the number of neurons compared to previous work, i.e. sub-exponential in the dimension instead of sub-linear for Daniely and Schacham [2020].
The proof is based on characterizing the landscape of the network function at different scales, showing that at relatively small "mesoscopic" scale of perturbations (o(sqrt(d)),  but still O(1)), the gradient is close to constant while being Omega(1), making it easy to find an adversarial example with a single gradient step.
Some experiments are provided to illustrate the theory, and to illustrate how the phenomenon extends to multiple layers.

**Limitations And Societal Impact:**

yes

**Main Review:**

The main result appears to significantly improve our understanding of adversarial examples for networks at random initialization, which have been crucial objects in the understanding of deep learning in recent years. The improvement compared to Daniely and Schacham seems significant, the paper is well written and the results seem novel, thus I recommend acceptance.

Here are a couple of minor comments:
* would the results easily extend to Gaussian initialization of a_l instead of +/- 1, which is perhaps more standard in practice?
* some comments on how these results could persist after initialization would be helpful (e.g. perhaps such lack of robustness could remain valid (only?) in the NTK regime?). Some experiments illustrating this on practical networks at various stages of training would also be welcome.



===== post-rebuttal update ======

Thank you for your response! Please do include your comments on Gaussian weights in the final version of the paper. I maintain my score.

**Time Spent Reviewing:**

2

---

> ### Author Response · Authors · 2021-08-10
> **Initial Author Response**
>
> Thank you for your comments!
>
> We believe our results to be easily extendable to the setting where the weights in the final layer are normally distributed. One simple way of obtaining these results is to first condition on the largest magnitude of the $a_\ell$. This is less than $C \sqrt{\log k / \gamma}$ with probability at least $1 - \gamma$. One can then carry out the rest of the proof conditioned on this event. This results in a weaker bound in Proposition 2 where the upper bound incurs an additional $\sqrt{\log k}$ multiplicative factor. Similarly, a lower bound on the size of the gradient follows by exploiting the anti-concentration properties of the Normal distribution using essentially the same proof as in Proposition 1. Here, one simply conditions on the event that at least $k/2$ of the $a_\ell$ have magnitude larger than $1/2$ which happens with exponentially high probability. Subsequently conditioning on the $a_\ell$ and carrying out the rest of the proof of Proposition 1 as is leads to a weaker bound with an additional multiplicative factor of (at least) $1 / 8$ in the lower bound. The upper bound on $|f(x)|$ follows a similar computation. We will include a more comprehensive discussion of this extension in the paper.
>
> The evolution of the landscape of the network through the training process is an interesting suggestion. While non-robustly trained networks still seem susceptible to one-step adversarial attacks, training procedures mitigating these effects have been designed. Understanding the effects of these is an important and intriguing direction for future work but is currently out of the scope of the paper.

---

### Official Review · Reviewer_aYEM · 2021-07-16

**Rating:** 7
**Confidence:** 4

**Summary:**

The authors prove that two-layer neural networks can be perturbed adversarially at a fixed data point and at random initialization (with high probability), i.e.\ for a neural network $f$ with randomly initialized parameters and a data point $x$, there exists a small perturbation $\delta = O(1)$ such that $|f(x)| = \Omega(1)$, but $f(x+\delta)\cdot f(x) < 0$. The result holds for a wide range of widths and activation functions. They present numerical evidence for their theoretical claims.

**Limitations And Societal Impact:**

the authors adequately addressed the limitations and potential negative societal impact of their work.

**Main Review:**

The results seem to be generally correct and are backed up by proofs and simulations. Personally, I find the observation that untrained shallow neural networks which are initialized randomly can easily be perturbed adversarially by a white box attack not that interesting. However, I believe that the results presented here are a significant improvement on e.g.\ Daniely and Schacham (2020), which was selected for NeurIPS 2020. Following precedent, I lean towards accepting the present work.

{\bf General comments:}

* The authors consider data $x$ in the sphere $\sqrt d \cdot S^{d-1}$, but construct adversarial examples in the ambient space $\R^d$. Is it possible to adapt the mechanics to adversarial examples on the data manifold? This could strengthen the authors' statement, but I am concerned that the hairy ball theorem may necessitate radial perturbations in high dimension. Can this be avoided (with high probability)?

* As is convention, the authors absorbed the bias $b$ in $\sigma(w\cdot x+b)$ into the weight vector $w$ by implicitly replacing a data point $x$ with a data point $(x,1)$. As they modify $x$ in the direction of the negative gradient, there is no reason for the last coordinate to be fixed, effectively changing the weights of the neural network rather than the data points. Could the authors address this discrepancy (either by more clearly stating the limitations of their result or by outlining a strategy to avoid this conflict). I believe that this could easily be avoided, for example by stepping in the direction of the partial gradient which does not take the last dimension into account.

* The authors name random variables $X, Y$ in the situation that $x$ is only used as a projection and the randomness stems from $w$ or $a$. I find this naming convention confusing and poorly chosen.

* The proofs are written with little detail, so that the reader has to reconstruct the argument. This makes a timely review different in a time-frame as short as NeurIPS, as it takes additional time to assess whether the details are correect. I would ask the authors to see if they can flesh out the details of the proofs in the appendix a little more.


{\bf Specific comments:}

* l 18: $|f(x)| \approx 1$ signals that $|f(x)| = 1 + o(1)$. I believe the authors mean that $|f(x)| \geq c>0$ for a constant which depends on $\psi$. Maybe $|f(x)| = \Omega(1)$ would be more appropriate.
* l 24: Typo 'Lipschtiz'
* l 37: Could the authors briefly explain the constraint $k\ll \exp(d^{0.24})$ and comment on the constant $0.24$? I do not follow exactly.
* l 67: The authors claim above that their main results hold for $\exp(d^\rho)$ with $\rho<1$, which is not the same as $\exp(o(d))$ (compare e.g.\ $\exp(d/\log d)$. A qualifier like 'almost', 'roughly' or 'essentially' might be appropriate. In the introduction (l 22) the same could be claimed, although there I do not see quite as large a problem.
* l 77: Does a continuous Gaussian process of this form admit adversarial examples?
* l 103: What does the term 'approximately distributed like' mean? Please be more precise.
* l 103: If $X, Y\sim N(0,1)$, then $\E[XY] \leq \sqrt{\E [X^2]\,\E[Y^2]} = 1$, but $\E[XY] = x\cdot u$ can be as large as $\Omega (\sqrt d)$. Please correct.
* l 103: The random variable $X$ is not related to the term in the previous line which contains the data point $x$ and induces the random variable $Y$. This notation is unfortunate and confusing.
* l 107: I assume that this would be $C^{2,1}$-smoothness, but please be more specific.
* eq (5): This claim is almost certainly wrong. It is much stronger than Lemma 5 in the appendix since an $\eps$-covering of the sphere $\sqrt d\cdot S^{d-1}$ requires $\sim (\eps^{-1} \sqrt d)^d$ elements. If $\|x\|$ is allowed to be polynomial in $d$, I do not believe this could be recovered. Please check this claim and provide a proof or find a different heuristic.
* l 116: A scalar-valued function which has constant height is automatically constant. Additionally, you are trying to argue that the function changes sign. This sentence is contradictory, and should be rewritten.
* l 148: Delete comma.
* l 163: Since the proof of Proposition 1 is given in the main text, it does not need to be listed in Appendix B.
* l 171: Typo 'Berstein'
* l 174: Where does the constant 10 come from? Also line 380.
* l 178: If the authors need more space for explanation, I believe the repetition of Theorem 1 is not necessary.
* l 248: The figures presented above suggest that $100,000$ was not reached, either.
* l 360: A few additional lines might make the proof easier to read (remark e.g.\ that $\delta$ is replaced by $\delta/2$, be clear about which variables Bernstein's inequality is used on, etc.)
* l 375: The Gaussians have variance $1/d$ and $\|\delta\|/d$ respectively, so the two factors should not appear with the same homogeneity (I believe the correct scaling would be $\|\delta\|^{q/2}$). Please double-check how this affects other results in the article.
* l 380: 'a union bound' (since English is weird). This also is not a union bound, but a tail inequality.
* eq (29): This looks vaguely correct, but I am slightly confused at the explicit expression for the first term. Could the authors explain more precisely which tail bound they are using? I would have expected $\log(1/\gamma)$ to be outside the square root in the first term.
* l 390: Since this was proven in the main text, it does not have to be repeated here.
* l 404: Please give more detail. Is there a factor 2 missing due to the absolute values?
* l 408: How does (11) enter? Please give more details.
* l 409: Typo 'crucial'
* l 417, 418: It looks like text versions do not line up
* l 420: The authors need to bound oscillations at an infinitesimal scale as well, but only {\em in terms of $\eps$}. This is different from estimating the Lipschitz-constant at scale $\eps$, which sounds more like taking $\sup_{\|x-y\|=\eps}\frac{\Phi(x) - \Phi(y)}{\eps}$.


**Time Spent Reviewing:**

5

---

> ### Author Response · Authors · 2021-08-10
> **Initial Author Resopnse**
>
> Thank you for your careful review and useful suggestions!
>
> Generating adversarial examples on the data manifold is an important consideration. In the $\mathrm{ReLU}$ setting, where the data manifold is the sphere $\sqrt{d} \mathbb{S}^{d - 1}$, a simple normalization suffices as the function computed by the network is positive homogeneous in the input and hence will still preserve the sign of the adversarial perturbation. In the smooth setting, it should suffice to consider the projection of the gradient orthogonal to $x$ and follow the same procedure. A more careful analysis of our proof yields $\|\mathcal{P}_x^\perp (\nabla f(x)) \| = \Omega (1)$ and $\| \nabla f(x) \| = O(1)$. Hence, the point $y = x + \eta \mathcal{P}_x^\perp (\nabla f(x))$ for $\eta = \tilde{O} (1)$ satisfies $\| y \| = \sqrt{d} + \tilde{O} (1 / \sqrt{d})$. Hence, the projection of $y$ onto $\sqrt{d} \mathbb{S}^{d - 1}$, $\tilde{y}$, satisfies $\| \tilde{y} - y \| \leq \tilde{O} (1 / \sqrt{d})$. Along with the fact that $\|\nabla f(x)\| = O(1)$, this establishes the fact that $\tilde{y}$ constitutes an adversarial example for $x$ on the manifold. Extensions to other data manifolds are an interesting open problem.
>
> The bias unit can also easily be incorporated into our results. First, let $V$ denote the subspace orthogonal to $e_d$; that is, the set of perturbations that do not alter the bias unit of the input. The conclusion of Lemma 2 in our paper can be modified to yield the stronger conclusion that $\|\mathcal{P}_V (\nabla f(x)) \| = \Omega(1)$ by changing the matrix $P$ used in the proof to the projection matrix onto the subspace orthogonal to $e_d$ and $x$. Hence, our conclusions hold even when adversarial perturbations are restricted to be in $V$ which by definition do not alter the bias unit. Hence, a bias unit can be incorporated into our results with a minor modification of our proof.
>
> We will revise the notation in the paper to avoid confusion over which variables appearing in our results are random. We will additionally include more comprehensive versions of our proofs in the appendix for the convenience of the reader.
>
> Specific comments:
> 1. l18: This is indeed incorrect. We will change this to denote $|f(x)| = O(1)$ as intended.
> 2. l37: Rearranging the third inequality in the statement of Theorem 2 shows that our results hold with probability at least $1 - 1 / \mathrm{poly} (d)$ as long as $k \leq \exp (d^\rho)$ for any $\rho < 1 / 4$. We will make this clearer.
> 3. l67: We will correct this. Thanks!
> 4. l77: The results of [Ben Arous et al, 2020] do yield adversarial examples for such Gaussian process. However, these results are limited to networks with polynomial activations and extensions to settings with even smooth activations seem technically challenging.
> 5. l103: We apologize for the confusion. This statement is meant to build intuition for the formal version of our proofs and the approximation is due to the application of the asymptotic central limit theorem to the random variable $u^\top \nabla^2 f(x) u$.
> 6. l103: We will change the notation of the random variables.
> 7. l107: Smoothness here is as in the statement of Theorem 1. We will make this statement more precise.
> 8. Equation 5: We apologize for the confusion. This heuristic was meant to build intuition for our final proof. The discussion preceding Equation 5 yields a weaker (heuristic) tail bound of $\sqrt{\log 1 / \gamma} / d$ on $v^\top \nabla^2 f(x) v$. A union bound over a net over the set $\\{(x, v): \|x\| \leq \mathrm{poly} (d) \text{ and } \|v\| \leq 1\\}$ yields the bound claimed in Equation 5. Equation 5 appears consistent with Lemma 5 which through a union bound over the set $\\{(x,\delta,v): \|x\| \leq \mathrm{poly} (d),\ \|\delta\| \leq R \text{ and } \|v\| \leq 1\\}$ yields the following weaker conclusion:
>
> $$
>         \forall x,y \in \mathbb{R}^d \text{ s.t } \\| x \\| \leq \mathrm{poly} (d) \text{ and } \\| y - x \\| \leq R: \|\nabla f(x) - \nabla f(y)\| \leq \tilde{O} \left(\frac{R}{\sqrt{d}}\right).
> $$
>
> Note that in both of the above sets, the log covering number is $\tilde{O} (d)$ yielding the required conclusions. We will improve the exposition around Equation 5 to make this clearer.
>
> 9. l116: We will change this to state that the height of the function is at most a constant.
> 10. l174: We apologize for the confusion. The constant $10$ is conservative choice of constant by an application of Corollary 4.2.13 from [Vershynin, 2018]. We will add the citation to the statement.
> 11. l178: Thanks for the suggestion! We will remove the restatement of the theorem if necessary.
> 12. l248: Indeed, the footnote was meant to explain why the x-axis had different ranges for various depths (the largest range being 1,000,000 for L=1). We will clarify this.
> 13. l360: We will improve the exposition of this section of the proof.
> 14. l375: The variances of the gaussians are $1 / d$ and $\|\delta\|^2 / d$ respectively. We will make this explicit.
> 15. l380: Here, we apply a union bound over a the $\epsilon$-net to the tail bound obtained in Lemma 5.
> 16. Equation 29: This manipulation follows from an application of Lemma 5 along with a union bound over the $\epsilon$-net. The term linear in $\log 1 / \gamma$ appears from the a product of the term outside the parentheses with the second term in the expression inside the parentheses.
> 17. l404: This expression appears correct without an additional factor of $2$. This bound follows from a bound on the pdf of a Gaussian random variable for the second term in the previous inequality and a standard Gaussian concentration inequality to the first. We will make this explicit.
> 18. l408: We apologize for the brevity. We have $\mathbb{P} (\\|w_\ell\\| \geq t / \epsilon) = \mathbb{P} (\\|w_\ell\\|^2 \geq t^2 / \epsilon^2)$. Since, $\|w_\ell\|^2$ is a sum of squared gaussian random variables, Equation 11 suffices to obtain a high probability bound on its value. We will make this explicit in our exposition.
> 19. l417/l418: We will fix the text here to avoid further confusion.
> 20. l420: We will correct this. Thanks!

---

### Official Review · Reviewer_2AAQ · 2021-07-17

**Rating:** 8
**Confidence:** 3

**Summary:**

The authors consider the existence of adversarial examples in neural networks trained by gradient descent.  They demonstrate that for two layer networks, one gradient update following random initialization can find adversarial examples.  Their results extend previous work by Daniely and Schacham to more general settings (for two layer networks) and rely upon a different suite of techniques.


**Limitations And Societal Impact:**

Adequately addressed

**Main Review:**

I quite like this paper.  They clearly explained the problem of interest, accurately described previous work and their limitations, and then proposed a solution to overcome the limitations of previous work.  The practice of finding adversarial examples using one-step attacks is common but had previously never been shown; this work thus provides a concrete theoretical explanation for this previously unexplained phenomenon.  I was unable to find any obvious drawbacks to the paper's results.

The exposition in the main section was particularly clear and nice to read.  In particular, the proof sketch was clear and intelligible, and the discussion in lines 72-87 about extensions to the width = exp(d) case was insightful.

The experiments could use some more details. e.g., which activation function is used? For Fig. 1, are the colored areas representing standard deviations over random initializations (if so how many)?

It may be helpful to emphasize in the introduction that the gradient is with respect to x rather than w (although this is implicit given your notation).

Is there any understanding of whether or not the results hold for more general distributions over x?  Some comments on this would be nice.


** Post rebuttal updates **
I've read the other reviewers' comments and the author response and am maintaining my strong recommendation for acceptance.  Thanks.

**Time Spent Reviewing:**

6

---

> ### Author Response · Authors · 2021-08-10
> **Initial Author Response**
>
> Thank you for your kind words! We will emphasize in the introduction that the gradients computed in the network are with respect to the input. As stated, our results hold for any $x$ chosen independently of the randomness in the network on $\sqrt{d} \mathbb{S}^{d - 1}$. However, due to the rotational invariance of the Gaussian distribution, our results also hold for a point chosen uniformly on the sphere. Additionally, for the $\mathrm{ReLU}$ activation, which satisfies the property of positive homogeneity, our results also hold for any distribution over $x$ as long as $\mathbb{P} \\{x = 0\\} = 0$. For the experiments, we used a ReLU activation. And indeed, the colored areas represent one standard deviation (as noted in the caption of the figure) over 100 random network initializations and 100 random input point per network. We will add details about these various points to the paper.

---

### Decision · Program_Chairs · 2021-09-27

**Decision:**

Accept (Spotlight)

**Comment:**

All reviewers find the paper to be clearly written, with a clear exposition of the problem and offering a novel analysis with a significant degree of novelty. I strongly recommend acceptance.